# Spiral packing and chiral selectivity in model membranes probed by phase-resolved sum-frequency generation microscopy

Alexander P. Fellows [1], Ben John [1], Martin Wolf [1] & Martin Thämer [1] ✉

Since the lipid raft model was developed at the end of the last century, it became clear that the specific molecular arrangements of phospholipid assemblies within a membrane have profound implications in a vast range of physiological functions. Studies of such condensed lipid islands in model systems using fluorescence and Brewster angle microscopies have shown a wide range of sizes and morphologies, with suggestions of substantial in-plane molecular anisotropy and mesoscopic structural chirality. Whilst these variations can significantly alter many membrane properties including its fluidity, permeability and molecular recognition, the details of the in-plane molecular orientations underlying these traits remain largely unknown. Here, we use phase-resolved sum-frequency generation microscopy on model membranes of mixed chirality phospholipid monolayers to fully determine the three-dimensional molecular structure of the constituent micron-scale condensed domains. We find that the domains possess curved molecular directionality with spiralling mesoscopic packing, where both the molecular and spiral turning directions depend on the lipid chirality, but form structures clearly deviating from mirror symmetry for different enantiomeric mixtures. This demonstrates strong enantioselectivity in the domain growth process and indicates fundamental thermodynamic differences between homo- and heterochiral membranes, which may be relevant in the evolution of homochirality in all living organisms.

The importance of phospholipids in biology cannot be overstated. Their amphiphilic nature promotes self-assembly into two-dimensional membranes where the hydrophobic tails tightly pack and point away from the membrane interface (see Fig. 1a)[1]. This well-defined molecular orientation normal to the membrane leaflet is the basis of their fundamental physiological purpose as barriers separating different environments within an organism. Biological membranes contain, among many other constituents, a complex mixture of saturated and unsaturated chiral lipids and form highly specialised regions that are involved in a plethora of physiological processes such as gas exchange mechanisms, endocytosis and exocytosis, cell signalling and cellular adhesion or recognition[2,3]. An example of such a specialised

region that has recently attracted much attention is lipid rafts, which are co-existing phases of ordered lipid islands (liquid-ordered, LO) within a disordered (liquid-disordered, LD) phase (see Fig. 1a)[4,5]. Lipid rafts have been linked to many of the aforementioned membrane functions but their exact nature and molecular structure are largely unknown, and their role in biological processes is highly debated[6–12].

One key to a better understanding of the formation, and ultimately the function of lipid rafts is encoded in their in-plane molecular structure. An important aspect of this structure is the amount of orientational correlation (order) that largely governs their physicochemical properties. The degree of in-plane order is connected to the lipid phase behaviour and thus the lateral interactions at the molecular scale.

[1]Fritz-Haber-Institute of the Max-Planck-Society, Berlin, Germany. ✉e-mail: thaemer@fhi-berlin.mpg.de

**Fig. 1 | Schematic structure of a phospholipid and different possible lateral packing structures. a** A schematic of a lipid bilayer membrane containing two lipids (red and purple headgroups) showing their well-defined out-of-plane directionality (black arrow) and ordered domains enriched in one lipid (red) surrounded by a disordered phase enriched in the other (purple). **b** A schematic of such ordered domains formed in a model lipid monolayer. **c** 3D representation of the (*R*)-DPPC lipid viewed from the side showing the two lipid tails [highlighted green (tail 1) and blue (tail 2)] on either side of a plane which cuts through the headgroup (highlighted in grey) and contains the defined in-plane direction (black arrow). This direction (as well as the highlighted plane) is defined as exactly opposite the in-plane component of the symmetric methyl stretch (CH₃SS) transition dipole (red arrows) for the two terminal methyl groups on the acyl tails, as demonstrated by projecting this component into the plane. The molecular stick structure of the tails highlights the terminal $CH_3$ groups in red and the $CH_2$ groups in green for emphasis. **d** 3D representation of the lipid viewed from the top, showing the separating plane (grey) containing the defined in-plane direction (black arrow) opposite to the CH₃SS in-plane component (red arrows). This top-view structure is also presented with a trapezoidal outline (blue, dashed line) as a representation of its asymmetric in-plane morphology. **e** Examples of possible in-plane packing arrangements, representing the lipid as its trapezoidal in-plane morphology (**d**) and in-plane direction (black arrow). CW clockwise, ACW anti-clockwise. Co- and counter-directional packing is defined by the specific adjacent lipid alignment directions.

Two limiting cases are the liquid-expanded (LE) phase, which shows no in-plane order, and the liquid-condensed (LC) phase, which displays long-range lateral crystallinity[13]. Whist these LC structures are thought to exist in certain classes of biomembranes such as the alveolar membrane in the lung, the lipid rafts formed in cell membranes are thought to adopt an LO phase with less orientational ordering. The extent and range of these correlations, however, are highly contentious. Beyond the pure in-plane order as a defining parameter, equally important for the properties of such ordered domains are the specific details of the underlying molecular packing and their inhomogeneities. Owing to the asymmetric in-plane shape of phospholipid molecules (Fig. 1c, d) there are various possibilities as to how they could be laterally assembled and arranged (examples are shown in Fig. 1e).

Which arrangements exist under which conditions is largely unknown due to the lack of microscopic insight. Furthermore, it is unclear to what extent lipid chirality influences the molecular arrangement within these ordered domains[14,15]. Enantioselectivity in the formation of lipid assemblies, resulting in different structures in homo- and heterochiral membranes, has been suggested amongst numerous tentative explanations for how and why evolution drove towards membrane homochirality in all branches of life[16–18]. Conclusive experimental evidence for such enantiomeric effects remains elusive. Clearly, a full molecular-level picture of the various types of physiological membranes would be desirable, however, such elucidation has proven difficult[8]. This fosters the importance of conceptual studies of these lipid ordering effects and the analysis of the lateral interactions in

model systems to shed light on the physics underlying the formation of such domains.

One of the most widely studied model systems for investigating the lateral arrangement and interactions in biomembranes is monolayers containing dipalmitoylphosphatidylcholine (DPPC), the most abundant phospholipid in cell membranes (Fig. 1b)[19–22]. Upon compression, using a Langmuir trough, monolayers of both pure DPPC and mixtures with other lipids (primarily unsaturated) form LC−LE coexistence equilibria with large DPPC-rich LC domains. These domains have been studied using various imaging techniques including fluorescence, Brewster angle, and atomic force microscopies[23–26]. These techniques allow for the visualisation of the LC domains and for investigating their morphologies. This has yielded a substantial insight into their growth and stability. For pure DPPC monolayers, for example, the formation of domains with spiralling morphologies has been observed, with clear mirror symmetry between the two enantiomers[27–29]. Such morphologies suggest a complex structural hierarchy possessing a well-ordered, crystalline molecular packing with a possible link between molecular and mesoscopic structural chirality. This interpretation has been supported by other experiments, e.g. X-ray diffraction measurements showing a high degree of in-plane order[30]. However, the details of the molecular packing in these lipid domains are inaccessible by such measurements and can only be inferred by their apparent morphologies. More detailed insight into their molecular structure has recently been gained using polarised fluorescence microscopy (PFM) which allows for probing the in-plane molecular ordering through the introduction of a fluorescent tag. Such studies have been performed on different types of phospholipid membranes including pure DPPC monolayers[20,31] and mixed lipid domains in bilayer systems[32–34]. The results have shown the presence of a generally curved in-plane molecular packing in various DPPC-rich domains and clearly give important initial insights into their structure at the molecular level. However, beyond these findings, many fundamental questions about the molecular structures in such phospholipid assemblies remain. This concerns the possible impact the tag might have on the observed structure and, more crucially, that the exact distribution of molecular orientations and thus the specific packing arrangement is still unknown. As PFM only yields the plane in which the tag molecules are aligned, the absolute molecular directionality cannot be obtained for two reasons. Firstly, only yielding the plane of orientation gives no indication of the specific direction within it, and secondly, the connection between the specific orientation and conformation of the tag and the lipid needs to be inferred. In consequence, important aspects of the molecular structures such as the co- and counter-direction arrangements (Fig. 1e) are indistinguishable based on PFM measurements. This impedes a detailed analysis of the intermolecular interactions which govern the membrane properties and prohibits us from drawing a clear connection between molecular chirality and the mesoscopic packing. Additionally, curved molecular packing has only been observed in highly asymmetric DPPC-rich domains. For DPPC monolayers mixed with unsaturated chiral lipids, the ordered domains are typically round,[35,36] which raises the question of whether these domains possess a similar molecular packing and whether this is a general feature of DPPC-rich domains. Additional open questions, concern the influence of other lipids on the growth and ultimately the specific internal packing within the domains, and whether there is any breaking of the mirror symmetry in such mixed systems with different enantiomeric combinations of the two lipids.

To address such questions, we developed a phase-sensitive vibrational sum-frequency generation (vSFG) microscope which can fully resolve the microscopic molecular structures of such lipid monolayers. In these experiments, we directly probe the phospholipid molecules through their C−H stretching vibrations. To investigate the molecular packing and interactions, we record hyperspectral images of different DPPC-rich domains in mixed monolayers of (R)/(S)-DPPC

and fully deuterated (unsaturated) 1-palmitoyl-2-oleoyl-glycero-3-phosphocholine ((R)-POPC) in a 4:1 ratio. This imaging technique combines spectroscopic selectivity to differentiate molecular species with sensitivity for absolute molecular orientation encoded in the phase of the signal, providing full molecular detail for these studies[37–42]. Whilst, such SFG microscopy measurements of molecular films on dielectric media have previously not been feasible due to technical challenges such as insufficient signal-to-noise ratios, our vSFG microscope employs an advanced imaging setup[43] that overcomes this limitation. This advancement not only allows the lipid monolayers to be visualised, but their specific packing arrangements and molecular structure to be elucidated at the sub-micron level.

## Results

In our vSFG experiments, the input and output beams are all set to P-polarisation (with their electric fields parallel to the $x'z'$-plane) and a 36° incidence angle to the surface normal ($z'$), implying significant excitation of molecular vibrations both parallel and perpendicular to the sample surface (see Fig. 2a, for further details, see 'Methods'). For lipid molecules, the generated vSFG signals in the C−H stretch region are known to almost exclusively originate from their alkyl tails[21]. As depicted in Fig. 1b, c, these tails are expected to be in a largely upright configuration with their terminal $CH_3$ groups all pointing upwards. For this structure, the vSFG responses should be dominated by the $CH_3$ groups and (with a 36° incidence angle) yield a positive contribution from the symmetric stretch (SS) and negative band from both antisymmetric (AS) modes (see Supplementary Note 5 for details). Furthermore, as the POPC is fully deuterated, only the vibrational resonances from DPPC are probed in these measurements.

### vSFG hyperspectral Imaging

The obtained vSFG images are presented in Fig. 2b, stepping through the vibrational spectrum at selected frequencies, with the spatially averaged spectrum shown in Fig. 2c (black, dashed line). Only the absorptive line shapes (imaginary parts of the phase-resolved responses) are presented, with Lorentzian-shaped peaks and dips centred at the frequencies of the vibrational transitions[44]. The first image, at $2815\,cm^{-1}$, does not correspond to any vibrational resonance from DPPC and thus no contrast can be seen. Thereafter, each image corresponds to one of the DPPC C−H resonances where the roughly circular domains become apparent. These domains have diameters of ca. $10\,\mu m$ and correspond to the DPPC-rich LC phase. Specifically, the images at $2875\,cm^{-1}$, $2935\,cm^{-1}$, and $2965\,cm^{-1}$ are associated with the $CH_3$ SS, its Fermi resonance (FR), and AS, respectively, showing strong positive (SS and FR) and negative (AS) responses, as expected[44]. These resonances also clearly feature in the averaged spectrum (with dashed lines indicating the frequencies associated with each image). Figure 2c also shows spectra averaged over only the observed domain regions (red), and only in the surrounding phase (blue). From this, it is clear that the majority of the vSFG signal arises from the DPPC-rich domains, as expected, but that the LE phase also contributes a significant signal. This reveals that the phase-separation of the two lipids is obviously incomplete, i.e. that both phases contain DPPC, only in differing proportions and likely with different molecular ordering.

The remaining two images, at $2845\,cm^{-1}$ and $2905\,cm^{-1}$, correspond to the $CH_2$ SS and its FR. For the well-packed upright lipid geometry, the $CH_2$ groups lie almost completely parallel to the surface and thus should mostly yield in-plane signals that are small due to the opposing directions of the $CH_2$ groups within each molecule[44]. Clearly, the $CH_2$ signals are much weaker cf. $CH_3$, following expectations. Interestingly, at the $CH_2$ frequencies, many of the domains also show a split positive/negative character. While the presence of $CH_2$ SFG signals in such monolayers is traditionally interpreted as arising from conformational (gauche) defects in the alkyl chains, these amplitude variations across the domains are clearly incompatible with the

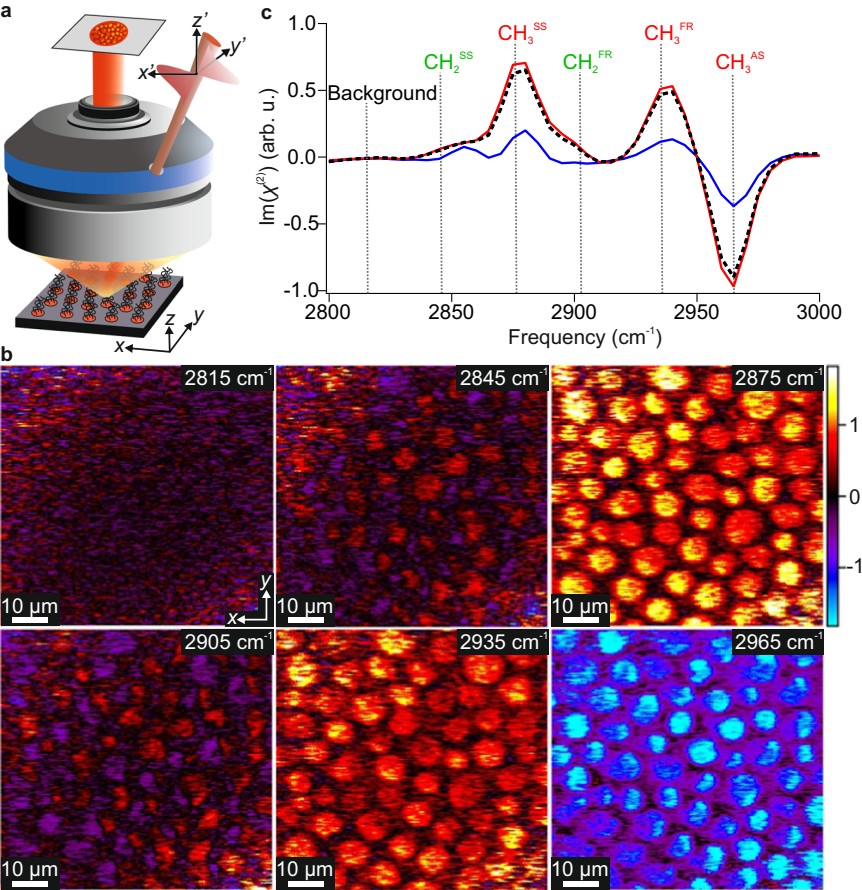

**Fig. 2 | vSFG microscopy images. a** Schematic of the phase-resolved vSFG microscope showing the input beams (red), lipid monolayer sample (similar to that shown in Fig. 1b) and the objective. Two coordinate systems are defined, the first ($x$, $y$, $z$) being the sample frame, and the second ($x'$, $y'$, $z'$) being the laboratory frame that is defined based on the plane of incidence of the input beams. **b** vSFG microscopy images of the ($R$)-DPPC/($R$)-POPC monolayer at select frequencies. The frequencies are given in the top-right of each image and correspond to the dashed lines shown in the spectra, and the image amplitudes all correspond to the imaginary part of $\chi^{(2)}$ given in arbitrary units and based on the colour bar shown on the right. **c** vSFG spectra averaged over all domains (red), outside the domains (blue) and the entire image (black, dashed line). Only the imaginary (absorptive) parts of the phase-resolved responses are shown. The main $CH_2$ and $CH_3$ vibrational resonances are also indicated (with $CH_3$ labels in red and $CH_2$ in green to coincide with the colours used in the molecular stick structure shown in Fig. 1c). SS symmetric stretch, AS antisymmetric stretch, FR Fermi resonance.

random nature of defects. Such defects would entirely cancel their in-plane contributions and only yield out-of-plane responses that should be largely homogeneous across the domains. Therefore, the dominant presence of both positive and negative contributions instead suggests highly correlated conformational structures that vary in directionality across the domains.

**Rotation-dependent vSFG imaging**

To unambiguously determine the molecular directionality, vSFG images are recorded as a function of sample rotation, yielding a four-dimensional dataset (vSFG images as a function of spectral frequency and azimuthal angle). The exact mathematical description of the rotationally dependent vSFG signals is highly involved due to the non-linear nature of the responses which contain multiple contributing susceptibility tensor components. Nevertheless, as discussed in Supplementary Note 3, they can be broken down and separated into an in-plane and out-of-plane contribution. Within this description, the rotation of the sample leaves the out-of-plane responses unchanged, whereas the in-plane responses are modulated sinusoidally. Therefore, by performing a Fourier transform (FT) of the rotations, thus converting them into azimuthal frequencies, the out-of-plane ($z'$) and in-plane ($x'$) components can be separated as they appear in different azimuthal frequencies. Specifically, the 0-fold frequency clearly must

only contain the out-of-plane response that remains constant, whereas the 1-fold frequency solely contains in-plane information that must change sign upon 180° rotation. A schematic of this analysis procedure is shown in Fig. 3a, b with further details and the underlying theoretical considerations of rotational vSFG responses given in Supplementary Notes 3 and 4. The resulting spatially averaged out-of-plane and in-plane spectra are given in Fig. 3c, d, respectively, along with their deconvolution (see 'Methods' for details) into resonances from $CH_2$ and $CH_3$, as well as two residual unassigned bands (shown in blue) which likely stem from either $CH_2$ groups from the alkyl tails or the head-group, or from the single CH (methine) group present in the head-group[21,44,45].

With the expected lipid orientation, the in-plane response could have signals from both $CH_2$ and $CH_3$ whereas the out-of-plane signal should be dominated by $CH_3$. Inspection of the deconvoluted spectra clearly shows this to be the case. Using these decompositions, we can also obtain both in-plane and out-of-plane magnitude images divided into $CH_3$ and $CH_2$ contributions (see Fig. 3e). These images confirm that the $CH_2$ groups yield strong in-plane and only minor out-of-plane signals, whereas the $CH_3$ groups give contributions to both, but are mostly present in the out-of-plane component. The sheer presence of in-plane SFG signals shows that the molecular packing is neither random nor counter-directional, but must be co-directional (Fig. 1e).

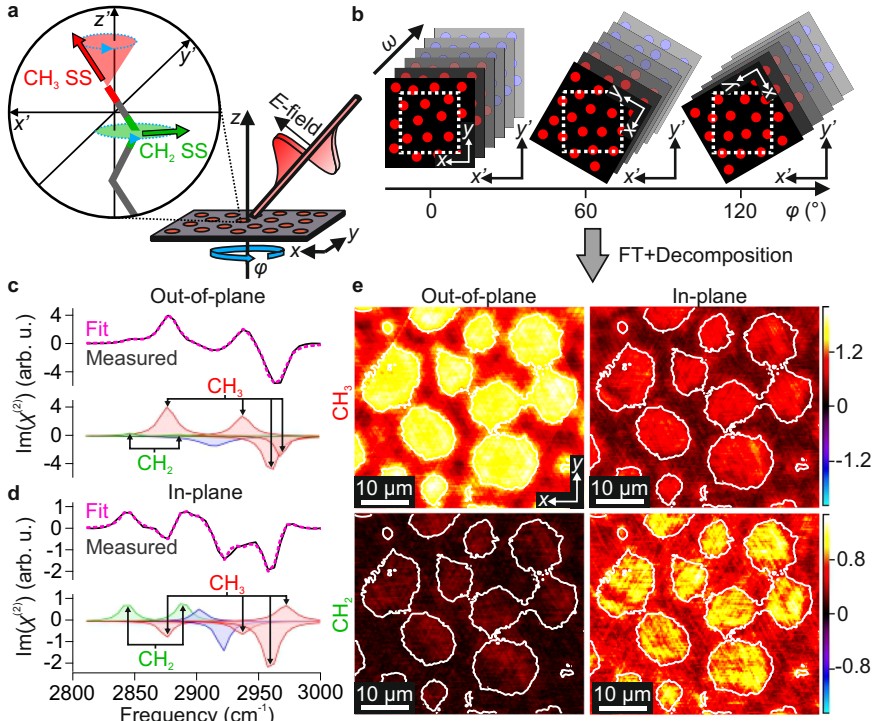

**Fig. 3 | Rotational analysis of vSFG microscopy images. a** Schematic of the rotational analysis procedure showing the variation in $CH_2$ (green) and $CH_3$ (red) transition dipoles upon sample rotation (indicated by blue arrows). **b** Schematic representations of the four-dimensional dataset that is generated, with SFG microscopy images at different spectral frequencies ($\omega$) recorded at multiple azimuthal sample rotations ($\varphi$). The change in sample frame coordinates ($x$, $y$, $z$) relative to the laboratory coordinates ($x'$, $y'$, $z'$) upon rotation is explicitly shown, demonstrating the need for back-rotation of the obtained images. **c** Imaginary part of the out-of-plane spectrum extracted from the magnitude of the 0-fold azimuthal frequency, averaged over all domain regions. **d** Imaginary part of the in-plane spectrum extracted from the magnitude of the 1-fold azimuthal frequency,

averaged over all domain regions. Also shown in the lower parts of panels **c**, **d** are the deconvolutions of the spectra into their constituent bands, with $CH_2$ resonances highlighted in green and labelled, $CH_3$ resonances in red and labelled, and unassigned bands in blue. **e** Magnitude images (in the sample frame) of the $CH_2$ and $CH_3$ resonances at the 0-fold (out-of-plane) and 1-fold (in-plane) azimuthal frequencies. The images are obtained by summing across the frequency regions of the $CH_2$ and $CH_3$ bands and are shown based on the colour scales given on the right, given in arbitrary units. White contour outlines based on the out-of-plane $CH_3$ amplitudes are included in all images to highlight the domain locations. The depicted surface region is part of a wider image common to all rotations, as shown in Fig. 4a.

This is clear as the inversion of molecular orientations inverts the sign of the response, and thus any in-plane inversion symmetry would lead to complete signal cancellation within each pixel.

With this data in hand, we can now fully determine the spatial distribution of both the out-of-plane and in-plane molecular orientations. As expected, the out-of-plane $CH_3$ signals (Fig. 3e) show a relatively homogeneous contribution for each domain which corresponds to all alkyl chains pointing up with a similar out-of-plane structure. Meanwhile, the in-plane molecular orientations are encoded in the rotational phases of the in-plane dataset (1-fold azimuthal frequency, see Supplementary Note 4), which are shown on the left side of Fig. 4b. The molecular axis that corresponds to these phases is defined according to the black arrows shown in Fig. 1c, d and again in the colour wheel in Fig. 4b (see Supplementary Notes 2 and 4 for details).

All the details about the distribution of molecular orientations are contained within the phase image, however, translating it into a clear structural picture requires additional analysis. In the first step, the molecular directions are extracted for selected regions and displayed as in-plane arrows, as shown in Fig. 4c (left side) for an exemplary domain. From this representation, it becomes immediately clear that the lipid packing is, indeed, curved. Since the absolute molecular orientations are obtained in these measurements, the curvature direction can be unambiguously determined as clockwise (CW) for the defined molecular axis.

To evaluate if there are any higher order hierarchal structural symmetries in these domains the phase contour lines connecting

locations of equal molecular orientation can be analysed. The left side of Fig. 4d shows a schematic of four possible CW curved arrangements, including both molecular direction arrows and the corresponding phase contours. Each structure creates characteristic contour line patterns (parallel, radial, or spiral with either positive or negative curvature directions) which can be used for their identification. Comparison to the contour lines extracted for the exemplary domain (also shown in Fig. 4c, left side) suggests that the mesoscopic structure of the domains is an outer subsection of a CW (+)-spiral. The formed domains thus possess a superposition of two types of curvature, namely the turning direction of molecules about the spiral centre, and the curvature of the spiral itself.

## Effect of molecular chirality

To further investigate the interplay between molecular chirality and mesoscopic structure in these systems, similar experiments with the other DPPC enantiomer (($S$)-DPPC/($R$)-POPC) are performed, with the results shown alongside the ($R$)-DPPC data in Fig. 4 (right side). For full comparability, identical preparation procedures and conditions are used for both samples. The obtained in-plane phase image (Fig. 4b) clearly differs for the ($S$)–($R$) mixture (cf. ($R$)–($R$)), suggesting different molecular structures. From the molecular direction arrows in the highlighted domain (Fig. 4c), it becomes evident that the ($S$)–($R$) mixture forms domains with similarly curved packing, but with an opposite curvature direction (anti-clockwise (ACW)). Meanwhile, the corresponding contour line image (also in Fig. 4c) shows that spiralling

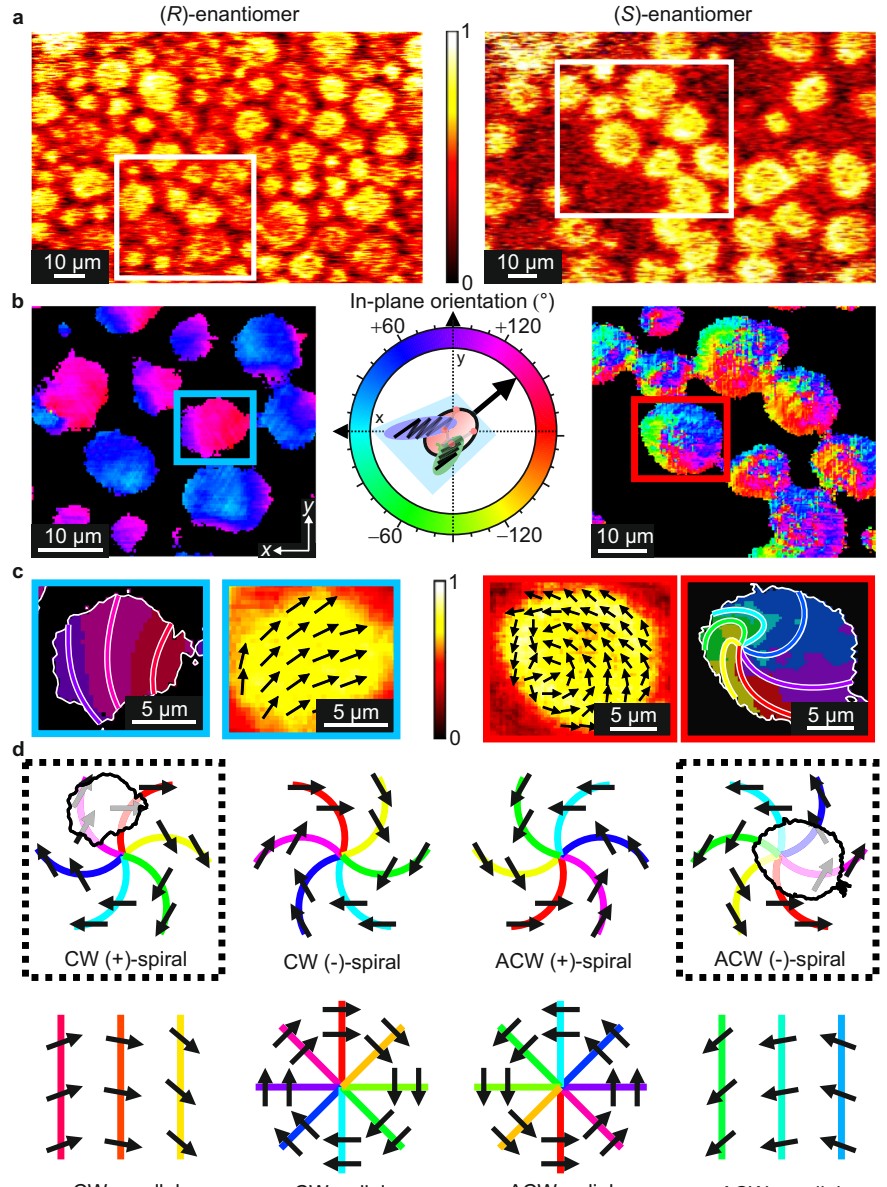

**Fig. 4 | In-plane phase directions of lipid rafts for each DPPC enantiomer.**
**a** Overview vSFG images showing the spectrally integrated CH$_3$ magnitudes for the two monolayer samples (left: (R)-DPPC/(R)-POPC and right: (S)-DPPC/(R)-POPC) at 0° rotation angle, based on the colour scale presented in arbitrary units. Also indicated are the surface regions common to all rotations (white box). **b** Rotational phase images from the in-plane component (1-fold rotational frequency) presented as colour maps (sample frame). The molecular directions are indicated in the phase colour wheel. **c** Magnified image of a selected domain from each phase map highlighting the distribution of molecular directions in the form of contour lines and phase arrows. Phase arrows are obtained by averaging 6 × 6-pixel regions across each domain and are shown on top of the normalised 0-fold magnitude image, given by the colour scale in the centre. **d** Schematic representations of different molecular packing structures, namely spirals (where '+' and '−' indicate the spiralling direction, defined as clockwise (CW) and anti-clockwise (ACW), respectively), parallel contours, and radial contours (concentric circles of lipids), with each showing the corresponding phase contour lines. Each structure is presented for both CW and ACW packing curvatures and the specific structures consistent with those observed in the highlighted domains (**c**) are indicated by outlining the structures with dashed boxes and overlaying the structure with the outline of the domains in a position roughly aligned with their observed phases.

mesoscopic structures are also formed, but with an inverted spiral direction (thus being part of an ACW (−)-spiral). This inversion of both types of curvature upon enantiomeric exchange confirms that the entire hierarchal structure is dominated by the DPPC chirality. It is important to note that, despite the obvious asymmetry between their domains, the underlying structural motifs of both mixtures are perfectly mirror symmetric. Interestingly, the observed asymmetry only originates from the location of the spiral centre with respect to the domain boundaries. While the (R)-enantiomer falls outside the domains, the (S)-enantiomer encloses them, making the 'full-spiral'

structure clearly visible. On inspection of the phase images in Fig. 4b, these structural properties are clearly common to all domains (see Supplementary Note 6 for further arrow representations). Interestingly, similar packing structures have been observed in various types of liquid crystals and superfluid helium droplets where the observed symmetry centres define an overall 'boojum' geometrical pattern[46–49].

Based on these findings, it can be concluded that the DPPC–POPC mixtures form condensed DPPC-rich domains circular in shape, but with curved and co-directional molecular spiral arrangements. Furthermore, alongside previous observations on pure DPPC

monolayers[27–29], this spirally molecular packing seems to be an intrinsic property of DPPC-rich LC domains. We find that both mixtures form domains based on mirror-symmetric spiral structures but represent different regions within such motifs (either excluding or enclosing their symmetry centres). This breaks their mirror symmetry. While the exact origin of this discrepancy cannot be assessed in the context of this work, this asymmetric structural difference between the domains formed in otherwise identical (R)–(R) and (S)–(R) lipid membranes is clearly a manifestation of enantioselective interactions between the two different lipids. The observed differences raise further important questions regarding protein binding and domain growth. Specifically, in the (R)–(R) mixture, the formed domains are structurally asymmetric (classified as a 'virtual' boojum with the symmetry centre falling outside the domain boundary), with one side showing concave packing and the other convex. In contrast, the full-spiral structures formed in the (S)–(R) mixture ('real' boojum) are somewhat rotationally symmetric but undoubtedly possess more substantial spatial heterogeneity. It would be interesting to study the impact of this difference on the location and strength of lateral binding to proteins[50–52]. Furthermore, these structures also imply that the domains in the (R)–(R) mixture possess varying line tensions around the domain circumference, whereas those in the (S)–(R) mixture are expected to be mostly constant. Such differences can be expressed in terms of free energies where the presence of a 'virtual' boojum ((R)–(R)) yields a more favourable smooth packing texture across the domain at a cost of non-tangential boundaries whereas a 'real' boojum ((S)–(R)) can minimise the line tension at the boundary but contains a higher energy internal packing. This further indicates substantive differences between the intermolecular interactions between each enantiomeric pairing. Since the line tension is generally an important factor in growth and aggregation/fusion[53–56], one can expect this to impact the formation, growth and resulting physicochemical properties of the domains in each enantiomeric mixture.

Experimental evidence for such differences in the formation and growth of the domains in the (R)–(R) and (S)–(R) mixtures can be found by further analysis of the vSFG images. On inspection of the wider images in Fig. 4a, it is clear that the (S)–(R) mixture possesses a lower surface coverage of the LC phase cf. (R)–(R). As detailed in Supplementary Note 7, we can quantitatively extract these coverages along with the vSFG signal magnitudes for the LC domains. This allows us to independently derive the relative molecular densities of each lipid within the domains and the degree of orientational order of DPPC. From this analysis, we find that the S–R domains possess ≈21% greater densities of DPPC, as well as ≈22% greater lipid order (i.e. narrower orientational distributions), and exhibit a higher DPPC excess. We can, therefore, conclude that the (S)–(R) mixture forms fewer LC domains which have a greater density and increased purity of DPPC, higher molecular order, and a hierarchal structure that includes the spiral centres, possibly suggesting slower formation kinetics. These observations can be further linked to the determined domain structures by considering the thermodynamic concepts of such boojum structures in liquid crystals. A greater molecular purity in the (S)–(R) domains suggests that the adhesive DPPC–POPC (hetero-molecular) interactions are less favourable for opposite chirality ((S)–(R)) compared to the same chirality ((R)–(R)). This hence indicates an overall greater line tension at the domain boundaries for the (S)–(R) mixture. Furthermore, the higher purity and DPPC density of the (S)–(R) domains show generally greater packing chirality. For liquid crystals, both of these effects favour the 'real' boojum geometric packing motif[46]. On the other hand, 'virtual' boojum structures are formed for generally lower line tensions and reduced packing chirality. Therefore, the extracted differences in domain purity and order clearly tally with the observed domain structures in both mixtures. Overall, this strongly implies that important physicochemical properties such as fluidity, stability and permeability, clearly differ between these homo- and heterochiral model membranes. Such substantial enantioselective effects could therefore be a relevant factor in the evolution of homochiral lipid membranes.

While important questions about the growth mechanism and general role of lipid rafts in biology obviously still remain unanswered, the determined hierarchal structure and observed breaking of mirror symmetry in the model systems investigated here represent a crucial insight into the nature of such assemblies and provide a detailed molecular picture of the ripened state within their formation process. To gain further understanding of the intermolecular interactions and lipid structures formed in biomembranes, it would be interesting to probe the domain packing structures and any enantioselectivity in the presence of other common membrane constituents such as cholesterol, phospholipids with charged head groups and different chain lengths, as well as proteins to assess how wide-spread, sensitive, and varied the lipid packing structures can be. Furthermore, an important evolution of such studies is the investigation of the formed structures in lipid bilayers. Whilst the lateral interactions mapped in monolayer models such as those investigated here are expected to dominate over the inter-leaflet interactions, it is not yet clear how significant any coupling is between the two leaflets in determining the specific lateral packing structure in trans-bilayer domains. This is of particular relevance for chiral lipids where only counter-directionality between the two inverted leaflets can maintain their individually preferred spiral packing directions. It is also worth mentioning that complementary information on the hierarchical chirality within such systems could be obtained through chiral SFG studies (as spectroscopy or even implemented in our microscope) as it can provide insight into chiral structures at the molecular level[57,58]. It would thus be interesting to extend these studies to specifically probe chiral tensor elements.

With the presented vSFG microscopy technique, a valuable tool becomes available for the investigation of such systems that can reveal their molecular structure in their full complexity, including in-plane orientations, a critical structural aspect that has previously not been accessible. Additionally, since the in-plane information is determined independently of any out-of-plane structure, investigations of bilayer systems such as those discussed above are entirely possible, despite the general inversion symmetry between the two leaflets. This advancement in microscopic vibrational imaging hence provides a promising perspective for further studies surrounding the significance of lipids rafts in our physiology.

## Methods
### Sample preparation
The lipids 1,2-dipalmitoyl-sn-glycero-3-phosphocholine ((R)-DPPC, or (L)-DPPC, >99%), 2,3-dipalmitoyl-sn-glycero-1-phosphocholine ((S)-DPPC or (D)-DPPC, >99%) and d82-1-palmitoyl-2-oleoyl-glycero-3-phosphocholine (d82-(R)-POPC or d82-(L)-POPC, >99%) are obtained from Avanti Polar Lipids (Alabaster, AL, USA), and dissolved in chloroform (99.0–99.4%, VWR International GmbH, Darmstadt, Germany) at a concentration of 1 mg ml⁻¹. Lipid mixtures are made by combining solutions in a 4:1 volume (mass) ratio of DPPC to POPC.

Mixed lipid monolayers are formed by depositing 20 μL aliquots onto the surface of a PTFE Langmuir–Blodgett trough (MicroTrough G1, Kibron, Helsinki, Finland) dropwise using an Eppendorf pipette. Prior to deposition, the trough is cleaned with both ultrapure water (Milli-Q, 18.2 MΩ cm, <3 ppb total organic carbon) and chloroform, filled with ultrapure water, and further cleaned via surface aspiration until the surface pressure was constant within 0.1 mN m⁻¹ at full compression. The platinum pressure sensor is flamed and rinsed with both chloroform and ultrapure water to remove any contaminants. Ultra-flat fused silica windows (Korth Kristalle, Altenholz, Germany, 5 mm thick, 25.4 mm diameter and <2 nm surface roughness) are rinsed with both chloroform and ultrapure water, exposed to UV-ozone (UV/Ozone ProCleaner Plus, BioForce Nanosciences Virginia Beach, VA, USA) for at

least 30 min, and dipped vertically into the sub-phase with a LayerX dipper (Kibron, Helsinki, Finland).

Once clean, with the fused silica substrate dipped, the lipid is deposited and left for a few minutes to let the chloroform evaporate. The film is then compressed to reach a constant pressure of 20 mN m⁻¹ with a barrier speed of 10 mm min⁻¹ and left further for ≈2 h to equilibrate and to allow the POPC lipid to oxidise due to the ambient ozone[59]. The monolayer is then cast by retracting the substrate through the interface at a speed of 2 mm min⁻¹.

### vSFG microscope
The developed heterodyned phase-resolved widefield vSFG microscope used in this work operates fully in the time domain, utilising the broadband IR and visible output from a Ti:sapphire-based, homebuilt interferometer, the details of which can be found elsewhere[60]. The IR (centred at 3450 nm) and visible (centred at 690 nm) beams are combined with a local oscillator generated from z-cut quartz in a collinear beam geometry. The three input beams, initially with ≈5 mm diameter, are focused by a 38 cm focal length off-axis parabolic mirror through a custom-drilled hole in a reflective objective (0.78 NA, Pike Technologies, Madison, WI, USA) at an incidence angle of 36° to reach the sample where the spot size is ≈100 µm. The sample is mounted on an automatic $XY$ rotation stage (SR50PP, Newport, Irvine, CA, USA) atop an automatic height controller (TRA25CC, Newport, Irvine, CA, USA) for full control of its position and orientation. The output beam from the objective is frequency-filtered to isolate the vSFG signals and recorded on an electrically cooled CCD camera (ProEM-HS:1024BX3, Teledyne Princeton Instruments, Trenton, NJ, USA) using paired-pixel balanced imaging (details can be found elsewhere[43]). In this work, the PPP polarisation combination is used such that both in-plane and out-of-plane components are probed. Specifically, with an incidence angle of 36°, the in-plane and out-of-plane electric field components are $E_{ip} = E_P \cos(36°)$ and $E_{op} = E_P \sin(36°)$, respectively. This hence yields relative in-plane and out-of-plane proportions as in Eqs. 1 and 2, showing ≈58% in-plane and ≈42% out-of-plane.

$$\frac{E_{ip}}{E_{ip} + E_{op}} = \frac{\cos(36°)}{\cos(36°) + \sin(36°)} = 0.579 \quad (1)$$

$$\frac{E_{op}}{E_{ip} + E_{op}} = \frac{\sin(36°)}{\cos(36°) + \sin(36°)} = 0.421 \quad (2)$$

### vSFG image treatment
vSFG images are obtained by acquiring interferometric images of the sample with 2 fs steps from −300 to 3000 fs time delay between the IR and other input beams (visible and local oscillator). Subsequent fast Fourier transform (FFT) of the result converts the interferogram images into complex spectral images.

The spectral images presented in this work represent the average of 4–12 individual interferometric scans (12 for the single images shown in Fig. 2, and at least 4 for each rotation for the data presented in Figs. 3 and 4). Phase and amplitudes of these raw spectral images are normalised by a reference vSFG microscopy measurement of a z-cut quartz crystal at reduced frequency resolution (scanning range from −300 to 300 fs). Each resulting image is then further treated by removing the dark counts, as well as a linear baseline (from non-resonant contributions) calculated based on points outside the spectral region of the resonant modes.

### Fitting and decomposition of vSFG spectra
The spatially averaged in-plane and out-of-plane spectra shown in Fig. 3c, d, as well as the second SVD component shown in Supplementary Fig. 1b, are deconvoluted into their constituent bands using an in-house Matlab fitting procedure. This fits the spectra with Lorentzian profiles based on the SFG equation[44] using a bounded non-linear least-squares curve fitting function with known estimates for the resonant frequencies of the relevant $CH_2$ and $CH_3$ bands from the literature[44]. Whilst the quantitative fits arising from such multi-parameter minimisation procedures often yield lower confidences, their purpose in this work is to qualitatively deconvolute the spectra to determine the relative contributions from $CH_2$ and $CH_3$ bands, and thus are more than adequate.

### Singular value decomposition (SVD)
As shown in the manuscript, the rotational analysis yields distinct in-plane and out-of-plane spectra that contain orientationally related contributions. To simplify the obtained 4D dataset and further improve the signal-to-noise ratio, a similar grouping of related spectral contributions can be obtained via principal component analysis. Here, this is performed via an SVD of the entire 4D dataset prior to the Fourier transformation of the rotational dimension (see Supplementary Note 1). As shown in Supplementary Figs. 1 and 2, the SVD yields only two significant components which highly match the obtained in-plane and out-of-plane spectra presented in Fig. 3c, d. Their in-plane and out-of-plane character is confirmed by comparing their 0-fold and 1-fold magnitude images obtained by Fourier transformation (see Supplementary Fig. 3). This analysis step hence both improves the data quality and allows the rotational phase determination to be performed on only a 3D dataset by extracting the in-plane component. The specific molecular direction of the extracted phase is then obtained using the signs of the specific transition dipole vectors in the in-plane SVD component spectrum.

The SVD analysis is performed by truncating the vSFG images along the frequency axis to only include the frequency range of interest, followed by a back-Fourier transform to the time domain to regenerate real (non-complex) data, and undertaking the in-built SVD algorithm in Igor Pro 8 (WaveMetrics, Lake Oswego, OR, USA). The output components are then converted back to the frequency domain via a further Fourier transformation.

### Rotational analysis
For the rotational analysis, vSFG images are recorded at sample rotations of at least every 60° across the full range. The treated images are then back-rotated and corresponding pixels are identified to overlap the images and create a 4D matrix of surface location (pixel) against spectral frequency and azimuthal (rotational) angle. After SVD, a new 4D matrix of surface location (pixel) against the SVD component and azimuthal angle is obtained which is converted into azimuthal frequencies via complex Fourier transform.

### Reporting summary
Further information on research design is available in the Nature Portfolio Reporting Summary linked to this article.

## Data availability
The data that support the findings of this study are available from the corresponding author upon request.

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

## Acknowledgements

We thank T. Khan, the Paarmann research group, M. Krenz along with the technician group, and the FHI mechanical workshop for their contributions towards the microscope design. We also thank A. Paarmann for help in editing the manuscript and the Bluhm research group for use of their LB-trough.

## Author contributions

A.P.F. and M.T. conceived the project and A.P.F., B.J. and M.T. designed the experiment. Samples were prepared by A.P.F. and B.J., and microscopy data were acquired by A.P.F., B.J. and M.T. The subsequent analysis was performed by A.P.F., B.J. and M.T., and all authors discussed the interpretation of the results. A.P.F. drafted the manuscript which was edited by all authors. M.T. and M.W. supervised the work and M.W. acquired funding.

## Funding

## Competing interests

The authors declare no competing interests.
