## [Peer Review File · Nature Communications]

Spiral packing and chiral selectivity in model membranes probed by phase-resolved sum-frequency generation microscopyREVIEWER COMMENTS

Reviewer #1 (Remarks to the Author):

This paper describes a new method of vibrational sum frequency generation microscopy to study the molecular tilt orientation in liquid condensed domains of DPPC in mixed saturated and unsaturated monolayer films. The films are transferred to silica windows via Langmuir-Blodgett deposition followed by a fairly complex spectroscopy/imaging technique developed by the authors. The main finding is that the tilt orientational ordering is different in right and left-handed DPPC; that is, the +1 disclination center is located within the domain for the non-natural S enantiomer and is external to the domain for the R enantiomer (also known as a "virtual boojum", Schwartz et al. *J. Chem. Phys.* 101, 8258–8261 (1994)). The authors speculate that the more uniform tilt orientation in the R enantiomer might be the origin of the enantiomeric selectivity of nature. The basis of this speculation is not well brought out.

There are some additional issues with the paper. The most important is that biomembranes do not really contain much, if any liquid condensed or LC phase, and rafts are not LC phase but are instead liquid ordered, LO, phases in liquid disordered, LD, phases, which are both liquid-like phases with no long range orientational ordering. LO-LD phases are distinguished by differences in cholesterol fraction. There are many papers, including those cited by the author, that make this distinction. LC-LE coexistence is likely in lung surfactants and tear films in the eye where there is a high concentration of saturated lipids.

A second issue is that visualizing tilt orientational ordering and effects of lipid type have been clearly visualized with polarized fluorescence microscopy in a series of papers: *J. Phys. Chem. Lett.* 2013, 4, 2789–2793; *J. AM. CHEM. SOC.* 2009, 131, 14130–14131; *Langmuir* 2014, 30, 10678–10685; *PNAS* 2013, vol. 110 3242–3247 and a nice paper on tilt boundaries in *Langmuir* 2015, 31, 9597–9601. It is necessary for the authors to put their work into context by considering all this work and its implications.

Overall, the authors have shown a nice new way of visualizing tilt orientation in LC phase domains, but they may be pushing their results a bit too far in their speculations regarding the origins of enantiomeric specificity.

Reviewer #2 (Remarks to the Author):

In this manuscript, Fellows et al. use phased-resolved sum-frequency generation microscopy to study the molecular structure of DPPC enantiomers (R/S-DPPC/R-POPC) at the interface. Using the azimuth angle analysis, the in-plane and out-of-plane contributions are extracted from SFG spectra. The in-plane contribution, which contains rotational phase information, revealed that both enantiomers have a spiral structure but opposite curvatures. As such, the distinct spiral structures indicate that the molecular chirality affects the lipid packing order.

The overall conclusion reached here, that molecular chirality affects the spiral structure, is in line with the conclusions from the earlier work (Ref 7: *Phys. Rev. E* 62, 7031–7043 (2000)). What seems new in this work is the novel azimuth angle analysis, which helps us to resolve in-plane/out-of-plane contributions. While the methodology seems promising, and I appreciate the authors' efforts on pushing the new imaging modality to extract molecular origins from the images, I found the explanation and interpretation of the data were mostly similar to how linear optical signal is analyzed, while here the optical signal is a 2nd order nonlinear signal. Thus, the current analysis lacks a bit of solid theoretical foundation, (e.g. a thorough analysis of rotation between lab and molecular frames to connect the second-order susceptibility in the lab frame, the molecular hyperpolarizability, and the tilt angles). This might be due to the special format of *Nat Comm* making the authors hesitant to go into more theoretical details. Nevertheless, further testing and a deeper exploration of the physical

meaning of in-plane/out-of-plane contributions would be beneficial to the community.

Comments:

In Figure 2 (b), the imaginary part of the SFG spectrum is very different with the spectrum at air-monolayer–solid interface (J. Phys. Chem. C 2016, 120, 15, 8175–8184). Especially, the phase is opposite. Since the phase of the imaginary component is related to molecular orientation, and both samples are an air-monolayer-solid interface, why is the phase opposite?

In line 107, the statement 'For this structure, the vSFG responses should be dominated by the CH3 groups and yield a positive contribution from the symmetric stretch (SS) and negative band from both antisymmetric (AS) modes, as indicated by the Z-components of their transition dipole vectors' requires additional explanation. In the ppp polarization combination, the SFG signal comes from four tensor elements contributions ($\chi_{xxz}, \chi_{xzx}, \chi_{zxx}$, and χ_{zzz}), for achiral systems (C_{∞}). More tensor elements become eligible with lower symmetry, such as C1 presented here, which was not included or discussed here. How can the sign of imaginary part directly connect to the direction of transition dipole vector?

Along the same line, the NA of the objective is not specifically large and the z component even at P polarization is not very large. Furthermore, depending on how large the laser beams are, the incidence angle can vary significantly. It is better to discuss these factors quantitatively, instead of claiming that there is a large z component of the incoming beam without quantitative descriptions.

In line 133, the statement 'Nevertheless, the simple presence of these CH2 responses 133 inside the domains shows that the signals from the individual lipids are not cancelling within each pixel, indicating substantial co-directional molecular in-plane ordering at the sub-micron scale.' requires additional explanation. The presence of CH2 response could also be attributed to the presence of gauche defect (Phys. Rev. Lett., 1987, 59, 1597–1600)

In the azimuth angle analysis, what is the physical interpretation of in-plane/out-of-plane component? It should be related to the tensor element of second-order nonlinear susceptibility, as well as the Euler angles (θ, ϕ, φ). The more detailed discussion should be included. For example, for a C1 symmetry component, all 2nd order susceptibility (and hyperpolarizability) are allowed, the current argument seems to be too hand-waving, and lack of a solid theoretical foundation.

In line 265, the authors calculated the lipid order and density based on several assumptions. However, the SFG response is the product of the molecular surface density (N), an order parameter (O), and the hyperpolarizability (β). The authors did not consider the hyperpolarizability in their analysis. The order parameter is typically determined by performing different polarization combination experiments in the SFG community. Therefore, it would be beneficial to conduct such experiments and discuss whether the results align with their current analysis.

In line 625, the statement 'an order parameter, O, that describes the width of the orientational distribution of the molecular transition dipoles' is incorrect. People usually assume different functions ($f(\theta)$) like exponential decay, Gaussian, or stepwise to describe the orientational distribution. The order parameter is the average of ensemble i.e. $\langle A \rangle = \int_0^\pi [Af(\theta)\sin\theta d\theta]$, $A = \cos\theta$ or $[\cos]^2 \theta$. See the references (J. Chem. Phys. 21 March 2006; 124 (11): 114705 and Phys. Rev. Lett. 121, 246101). As such, the order parameter cannot provide information about the width of the orientational distribution.

In the chirality study, people usually use chiral SFG spectroscopy for characterizations of molecular structures at interfaces (Chem. Rev. 2014, 114, 17, 8471–8498). However, there is no any discussion

of chiral SFG in the manuscript. If the author performed the chiral SFG experiment, could they get the consistent result as their current analysis? The discussion and references about chiral SFG should be included.

Lastly, there could be two types of chiral signals, one coming from the chirality of the molecules themselves, another coming from a chiral structure arranged by molecules (either chiral or achiral) in a specific way. It seems that the authors could see both of them. It would be helpful if the authors can discuss more about the origin of the signals.

In summary, the paper presents the new methodology and data for characterizing the lipid domain with mixed chirality. The conclusions reached are broadly in line with earlier work. The discussion in some places is lack of necessary details to establish a solid foundation to appreciate the data obtained from this advanced technique (see above). A more specialized journal might allow for a more rigorous discussion.

Reviewer #3 (Remarks to the Author):

In this manuscript, Fellows et al. describe a new microscopy technique that allow the visualization and study of lipid rafts in lipid monolayers. The authors show that by probing the C-H stretching vibrations of DPPC domains it is possible to resolve the microscopic structure of lipid monolayers. The paper is well-written and the results are well-supported by the experiments. I recommend its publication in Nat. Commun. after addressing the following comments:

- My main concern is related to the clarity of the in-plane molecular directionality. While the out-of-plane directionality is clearly explained and represented, the in-plane directionality of individual molecules, which is central to this work, is not clearly explained and it is hard to visualize despite the expansion present in Figure 1a. It is not clear what the black and blue arrow indicates. Since the in-plane directionality derives from the chirality, a 3D representation of the phospholipid molecule would help clarify this concept and make the manuscript more accessible to a broader readership.

- The developed microcopy technique seems very powerful to characterize lipid monolayers but lipid bilayers are never mentioned throughout the manuscript. This is a limitation that should be mentioned and discussed. If the technique is applicable only to monolayers, this should be reflected in the title of the paper.

- The DPPC/DOPC ratio used to obtain lipid rafts should be mentioned at least once in the "Results" section.

- The observation that DPPC is present both in liquid-condensed (LC) and liquid expanded (LE) phases is very interesting. Would it be possible to estimate their proportion? If yes, given a specific DPPC/POPC ratio, this would allow to study and what factors influence the formation of lipid rafts.

- The findings summarized with the sentence "We find that both mixtures form domains based on mirror-symmetric spiral structures, but represent different regions within such motifs. This breaks their mirror symmetry" should be better described and discussed. The authors should explain better the meaning of breaking the symmetry and from what evidence these findings are derived.

Reviewer #4 (Remarks to the Author):

I co-reviewed this manuscript with one of the reviewers who provided the listed reports. This is part of the Nature Communications initiative to facilitate training in peer review and to provide appropriate

recognition for Early Career Researchers who co-review manuscripts.

Reviewer #5 (Remarks to the Author):

Reviewer #6 (Remarks to the Author):

Response to Reviewers' Comments

Reviewer #1

This paper describes a new method of vibrational sum frequency generation microscopy to study the molecular tilt orientation in liquid condensed domains of DPPC in mixed saturated and unsaturated monolayer films. The films are transferred to silica windows via Langmuir-Blodgett deposition followed by a fairly complex spectroscopy/imaging technique developed by the authors. The main finding is that the tilt orientational ordering is different in right and left-handed DPPC; that is, the +1 disclination center is located within the domain for the non-natural S enantiomer and is external to the domain for the R enantiomer (also known as a "virtual boojum", Schwartz et al. J. Chem. Phys. 101, 8258–8261 (1994)). The authors speculate that the more uniform tilt orientation in the R enantiomer might be the origin of the enantiomeric selectivity of nature. The basis of this speculation is not well brought out.

We thank the reviewer for providing us with their expert review of our manuscript. For each of the points they have raised, a response is given below and the changes to the manuscript described.

Firstly, regarding the speculation surrounding the importance of the observed enantioselectivity in the domains of the two lipid mixtures, whilst our results do clearly show that such enantioselectivity must exist and therefore represent enantioselective interactions between the two lipids that alter the thermodynamics of the domain growth process, we concede that too much emphasis is put on this finding in relation to the impact it may have on the origin of enantioselectivity in nature. We have therefore altered the discussion of this point to more precisely describe what we find and how this may give some insight into the enantioselective interactions within membranes. We do maintain, however, that such thermodynamic differences between R-R and R-S mixtures have been discussed amongst suggestions for why membranes are entirely homochiral since the last universal common ancestor (LUCA). For this reason, we still mention this in relation to our findings, only with more speculation surrounding its significance. Nevertheless, we have rather extended the discussion on the thermodynamic interpretation of the results in terms of the lipid packing interactions and forming the different types of boojum defects.

There are some additional issues with the paper. The most important is that biomembranes do not really contain much, if any liquid condensed or LC phase, and rafts are not LC phase but are instead liquid ordered, LO, phases in liquid disordered, LD, phases, which are both liquid-like phases with no long range orientational ordering. LO-LD phases are distinguished by differences in cholesterol fraction. There are many papers, including those cited by the author, that make this distinction. LC-LE coexistence is likely in lung surfactants and tear films in the eye where there is a high concentration of saturated lipids.

The reviewer raises a very important point here with the specific structures formed in different lipid mixtures within a membrane. Whilst we completely agree that the presence of cholesterol in membranes does importantly alter the packing structure within lipid domains, leading to more fluid-like behaviour, we believe that suggesting that no long-range molecular order exists in these systems is somewhat contentious. The ordered nature of these systems (which is clear from the many SFG spectroscopic studies on cholesterol-rich membranes) implies that the lateral interactions between lipid tails is large. It is hence reasonable to suggest that they can

also display long-range structural correlations. This has, for example, been suggested previously elsewhere. [Soft Matter, 2009, 5, 3174–3186] We maintain, therefore, that the study of lateral lipid packing in model membranes (as in this work) is important for the understanding of many structural aspects of all biomembranes at the conceptual level despite the differences that they may show with other constituents present. In fact, we believe that our SFG microscopy technique showcased here represents a powerful tool to address these questions in the future. We would also like to note that several papers, including those listed by the reviewer in the following point which are published in high quality peer-reviewed journals, utilise other techniques such as PFM to study lipid interactions in similar systems of phospholipid membranes in the LC (gel) phase and clearly also emphasise the importance of studying these systems for the elucidation of lipid interactions in biomembranes.

As this point was raised by the reviewer, we have made several alterations to the manuscript to address it. Firstly, we have included more discussion on the different packing structures and lipid phase behaviour along with their relevance to biomembranes on page 2. Secondly, we have altered Figure 1 so as to not describe the domains present in the schematic bilayer as “condensed” domains but rather as “ordered” domains. Thirdly, we have put more emphasis on page 2 and 3 to the context of studying these model systems in order to conceptually understand lateral lipid interactions, across all types of biomembranes. Finally, we have added to the discussion of the results and outlook of this work on page 9 and 10 to discuss the broader relevance of our findings in terms of understanding lipid membrane structures along with the potential of this technique to further address questions such as the impact of cholesterol on lateral lipid ordering amongst others.

A second issue is that visualizing tilt orientational ordering and effects of lipid type have been clearly visualized with polarized fluorescence microscopy in a series of papers: J. Phys. Chem. Lett. 2013, 4, 2789–2793; J. AM. CHEM. SOC. 2009, 131, 14130–14131; Langmuir 2014, 30, 10678–10685; PNAS 2013, vol. 110 3242–3247 and a nice paper on tilt boundaries in Langmuir 2015, 31, 9597–9601. It is necessary for the authors to put their work into context by considering all this work and its implications.

We agree that polarised fluorescence microscopy studies of lipid membranes such as those highlighted by the reviewer represent an important series of investigations for the characterisation of lipid interactions and packing structures within biomembranes, and acknowledge that the lateral lipid ordering has indeed been clearly demonstrated by this technique. Nevertheless, the presented results employing SFG microscopy are clearly not equivalent to the information obtained by PFM.

Firstly, whilst PFM can determine the orientational plane of ordering within the membrane, it cannot yield absolute directions. By contrast, as we mention in the paper, the phase sensitive SFG measurements are highly sensitive to specific directionality. This ability to obtain absolute molecular directions is crucial for characterising the domain structures, as demonstrated by the schematic structural motifs in Figure 4. Whilst obtaining the orientational ordering plane can distinguish between hierarchical structures with different spiralling directions i.e., (+)- or (-)-spirals, it cannot distinguish between the two classes of each based on the molecular directionality i.e. ACW vs CW. As the molecular directionality yields information about the specific molecular growth mechanism of the domains, this information represents an important aspect of the understanding of lipid interactions within biomembranes. Furthermore, without sensitivity to the absolute directionality, it is impossible to distinguish between co- and counter-

directional packing (schematically shown in Figure 1). Clearly, parallel and antiparallel alignment can exist in elsewhere in biological systems, with an obvious example being β -sheet protein structures, and thus this structural property represents another important feature of the lipid packing. Access to this feature is nevertheless not possible with PFM and, to the best of our knowledge, phase-resolve SFG microscopy is currently the only technique capable of revealing these details.

Secondly, PFM measurements require the use of fluorescent tags which clearly have strong interactions with the lipid molecules as substantial order in them is induced within the condensed domains. Whilst only present in relatively small concentrations, these strong interactions clearly indicate a perturbation to the system which may alter the lateral interactions within the membrane. This is clearly the case with several membrane-associated molecules, particularly those which insert themselves within the membrane such as cholesterol. Therefore, when probing such samples with PFM, the question about how the tag influences the structure is always present. For our SFG microscopy measurements, however, no tags are required and thus the native structure is directly probed, without any potential molecule-induced perturbations.

Thirdly, as the PFM measurements probe the tag rather than the lipid, they represent an indirect measure of the system and thus the specific relation between the orientation of the lipid and the tag presents a further obstacle for characterising the lipid packing structure. In contrast, SFG directly probes the lipid vibrations and thus gains direct access to the molecular directionality. Finally, a further benefit of SFG compared to PFM is that it can gain access to molecular recognition in combination with the characterisation of their 3D molecular orientation and specific conformations. For example, in this work, only the DPPC lipid contributes to the observed signals as the POPC is deuterated. However, even if both are protonated, their subtly different spectra and the conformational difference associated with having differing tail-group functionalities would allow for them to be distinguished.

Furthermore, in addition to the in-plane orientational information at the heart of this work, given that the out-of-plane information can also be obtained (despite being more trivial in this case with regards to the lipid packing structure), a full 3D structure of the lipid domains can be determined. The presented study represents therefore an important step forward in the investigation of such monolayers as it gives access to crucial information that has been so far inaccessible. Despite all these advantages, however, PFM still represents an important technique, particularly for the study of kinetic processes or of dynamical systems thanks to its clear advantage in acquisition time. In this sense, the two techniques can be considered as complementary characterization tools.

As this point was raised by the reviewer, we have added substantial commentary on the capabilities and of PFM and the type of information that is and has been accessible through its application to lipid membranes. Additionally, we have placed more emphasis on the importance on the remaining open questions relating to structural aspects that cannot be addressed by PFM studies. This can be found on page 4.

Overall, the authors have shown a nice new way of visualizing tilt orientation in LC phase domains, but they may be pushing their results a bit too far in their speculations regarding the origins of enantiomeric specificity.

Once again, we thank the reviewer for their comments and hope that our detailed responses along with the alterations to the manuscript satisfy their concerns, particularly that the interpretations of the results are not overly speculative and that the technical advancement showcased in this study yields both important new insights into the structural interactions between lipids in biomembranes and shows substantial potential for future studies on structural heterogeneity in molecular films beyond what is currently possible through other methods.

Reviewer #2

In this manuscript, Fellows et al. use phased-resolved sum-frequency generation microscopy to study the molecular structure of DPPC enantiomers (R/S-DPPC/R-POPC) at the interface. Using the azimuth angle analysis, the in-plane and out-of-plane contributions are extracted from SFG spectra. The in-plane contribution, which contains rotational phase information, revealed that both enantiomers have a spiral structure but opposite curvatures. As such, the distinct spiral structures indicate that the molecular chirality affects the lipid packing order.

The overall conclusion reached here, that molecular chirality affects the spiral structure, is in line with the conclusions from the earlier work (Ref 7: Phys. Rev. E 62, 7031–7043 (2000)). What seems new in this work is the novel azimuth angle analysis, which helps us to resolve in-plane/out-of-plane contributions. While the methodology seems promising, and I appreciate the authors' efforts on pushing the new imaging modality to extract molecular origins from the images, I found the explanation and interpretation of the data were mostly similar to how linear optical signal is analyzed, while here the optical signal is an 2nd order nonlinear signal. Thus, the current analysis is lack a bit of solid theoretical foundation, (e.g. a thorough analysis of rotation between lab and molecular frames to connect the second-order susceptibility in the lab frame, the molecular hyperpolarizability, and the tilt angles). This might be due to the special format of Nat Comm making the authors hesitant to go into more theoretical details. Nevertheless, further testing and a deeper exploration of the physical meaning of in-plane/out-of-plane contributions would be beneficial to the community.

We thank the reviewer for their time in reviewing our paper and providing us with a detailed critique. Each point raised by the reviewer has been directly commented on and, where relevant, the details of the alterations within the manuscript are given. Whilst the reviewer clearly recognises the novelty of the rotational analysis methodology for obtaining molecular directionality in SFG microscopy, we would like to emphasise that the novelty of this study extends far beyond this. Firstly, SFG hyperspectral images of molecular films on a dielectric medium has thus far never been achieved. Here, not only do we show this for the first time, but with the rotational analysis methodology, we go significantly further by reaching an advanced level of experimental analysis to gain unprecedented insight into these systems. Specifically, to the best of our knowledge, the absolute directionality in molecular monolayers has never been extracted experimentally. As such, we have placed more emphasis on the importance of this technical development on page 4.

It seems like the main concern of the reviewer is associated with the theoretical basis of the rotational analysis procedure in determining molecular directionality. As the reviewer recognised, the general lack of in-depth theoretical descriptions in the manuscript is due to the format and style of Nature Comm. articles. We can confirm that the second-order nature of the optical responses have been fully included and the presented findings are based on far more detailed analysis and additional confirmatory measurements than shown in the main text. Overall, we believe that the novelty of the technique along with its insights into the lipid packing structure are of significant interest to a wider community such as the readership of Nature Comm. and thus that this necessitates good readability and accessibility in this paper, which would not be possible with a substantial theoretical description of the technique. Clearly, however, we have gone too far with the condensation of the theoretical description in the paper in its current version, which could indeed appear as being oversimplified. Nevertheless, we plan to publish a highly technical paper detailing these in full at some point soon in a specialised

journal. In the meantime, we hope that the following description will reassure the reviewer that the presented results and analysis are on a solid theoretical foundation. Additionally, despite the theory of second-order responses already existing elsewhere in the literature, as this concern arose, we have added to the manuscript on page 6 to give more insight into the second-order nature of the responses and how this leads to the orientational information in addition to a further description of the fundamental theory underlying the method in the Supplementary Information.

As the reviewer noted in their comments below, the responses in PPP polarisation are generally contributed to by 8 tensor components, of which only 4 are relevant when probing an azimuthally symmetric (C_{∞}) system. For the heterogeneous C_1 symmetry which we show to be clearly present in the sample systems probed in this work, however, all 8 can contribute, namely: XXX , XXZ , XZX , ZXX , XZZ , ZXZ , ZZX , and ZZZ . With a collinear beam geometry, as employed here, however, 4 of these contributions effectively cancel in the overall response, namely XZX cancels with ZXX , and XZZ cancels with ZXZ . This leaves just the XXX , XXZ , ZZX , and ZZZ contributions.

By considering the susceptibility contributions in the sample frame, each tensor component obviously follows a cubic spatial symmetry such that rotation of the sample around the Z axis (ϕ) modulates their contributions as follows:

$$XXX \sim \cos^3(\phi)$$

$$XXZ \sim \cos^2(\phi)$$

$$ZZX \sim \cos(\phi)$$

$$ZZZ \sim 1$$

As the sample is rotated about the Z-axis, further components become accessible by interchanging X with Y, namely:

$$YYY \sim \sin^3(\phi)$$

$$YYZ \sim \sin^2(\phi)$$

$$ZZY \sim \sin(\phi)$$

Through the trigonometric equalities, after Fourier transformation of the rotational measurements, this means the XXX and YYY components will have contributions in the 1-fold and 3-fold rotational frequencies, XXZ and YYZ in the 0-fold and 2-fold, ZZX and ZZY in the 1-fold, and ZZZ in the 0-fold. Hence, by looking at the 0-fold and 1-fold frequencies, the former contains contributions from XXZ , YYZ , and ZZZ , whereas the latter contains contributions from XXX , YYY , ZZX , and ZZY . It is important to note that the 1-fold frequency thus only contains contributions probing the resonant X and Y components (IR) whereas the 0-fold only contains resonant Z components. Clearly, therefore, this allows the overall response to be split into an in-plane and out-of-plane contribution. Additionally, as the in-plane component is defined only by the 1-fold rotational frequency, it MUST only contain contributions which oscillate as pure sine waves, i.e. any higher order rotational contributions have been filtered out. Therefore, it logically follows that the phase of the in-plane contribution must be directly and monotonically connected to the in-plane molecular directionality.

In fact, as shown in the added theory section in the Supplementary Information, by explicitly considering the molecular hyperpolarisabilities and their projection onto the lab-frame coordinates, the phase of the 1-fold in-plane response is exactly equivalent to the in-plane Euler angle of the transition dipole for the CH₃SS. Hence, as this is the basis for the extraction of the molecular directionality in the paper, this demonstrates that such a treatment, whilst appearing as only for linear responses, is completely valid for such non-linear signals.

As a final note, for further certainty, we have also performed a similar analysis in the SSS polarisation combination which solely probes the in-plane components: YYY and XXX (with rotation about Z). Again, by relating this contribution to the molecular hyperpolarisabilities, it can also be shown that the phase of the 1-fold response equates to the molecular directionality. The obtained molecular directions from the SSS response were found to be in good agreement with those from PPP measurements, thus lending further confidence in the presented results.

Clearly the theoretical details underlying this analysis and mentioned additional confirmatory measurements are highly substantive and thus go far beyond the scope of this publication targeting a broad readership. Nevertheless, we have added more information to both the main text and Supplementary Information to widen the theoretical basis of our approach, as requested by the reviewer.

In Figure 2 (b), the imaginary part of the SFG spectrum is very different with the spectrum at air–monolayer–solid interface (J. Phys Chem. C 2016, 120, 15, 8175–8184). Especially, the phase is opposite. Since the phase of the imaginary component is related to molecular orientation, and both samples are an air-monolayer-solid interface, why is the phase opposite?

In this work, the phases are determined relative to z-cut quartz as detailed in the publication [J. Chem. Phys. 151, 064707 (2019)]. Whilst we cannot comment directly on how the phase of the spectrum presented in the reference mentioned by the reviewer was determined, we acknowledge that it appears to be opposite. It is important to note, however, that correct phase referencing can be challenging, with other examples of SFG spectra in the literature having opposite phases for the same system being probed. [Phys. Chem. Chem. Phys., 2021, 23, 18253–18267] In any case, we can further confirm that the phase of our presented spectrum is correct by considering the PPP spectra of SAMs on gold (which is effectively only the ZZZ contribution) and the SSP spectrum of a lipid monolayer (which probes YYZ=XXZ in azimuthally symmetric systems). The ZZZ contribution appears solely as dips for the three methyl resonances (given that gold is well-known to have a positive phase for its NR contribution) whereas the YYZ (XXZ) appears as dips for the SS and FR, but a peak for the AS. Therefore, as the spatially averaged PPP spectrum is effectively given by ZZZ – XXZ, this leads to cancellation of intensity for the SS and FR and a strong negative band for the AS. This describes the appearance of PPP spectra elsewhere within the literature of monolayer lipid systems well, with generally weak SS and FR and a much stronger AS. Clearly, the AS in our presented spectra is negative, thus aligning with this description. It is also worth noting that the phase we present here has also been reproduced elsewhere for PPP spectra when referencing to the +x z-cut quartz response. [Biointerphases 18, 058502 (2023)] The only other difference with our presented spectra arises from the relatively small incidence angle of 36° employed here, contrasting with most other systems, making the XXZ contribution relatively more significant, and thus not yielding the same cancellation of the SS and FR as seen elsewhere in the literature. Therefore, we are convinced that both the phase and appearance of our spectrum is correct. As this comment was made, we noticed that the angle of incidence used in this work

was only given in the Supplementary Information and have thus added it into the main manuscript on page 4 and the methods section as this is crucial to the understanding of why the spectra appear as they do.

In line 107, the statement ‘For this structure, the vSFG responses should be dominated by the CH3 groups and yield a positive contribution from the symmetric stretch (SS) and negative band from both antisymmetric (AS) modes, as indicated by the Z-components of their transition dipole vectors’ requires additional explanation. In the ppp polarization combination, the SFG signal comes from four tensor elements contributions (χ_{xxz} , χ_{xzx} , χ_{zxx} , and χ_{zzz}), for achiral systems (C_{∞}). More tensor elements become eligible with lower symmetry, such as C_1 presented here, which was not included or discussed here. How can the sign of imaginary part directly connect to the direction of transition dipole vector?

We agree with the reviewer that this statement is over-simplifying the situation, although would like to point out that there is a well-defined link between the molecular orientation and the sign of its hyperpolarisability components, even if this becomes less well-defined when combining the ZZZ and XXZ contributions that dominate the observed spatially-averaged SFG spectra presented in Figure 2. We have modified the manuscript to remove this somewhat misleading statement as it has no consequences for the interpretation of the presented data.

Along the same line, the NA of the objective is not specifically large and the z component even at P polarization is not very large. Furthermore, depending on how large the laser beams are, the incidence angle can vary significantly. It is better to discuss these factors quantitatively, instead of claiming that there is a large z component of the incoming beam without quantitative descriptions.

Whilst we somewhat contest that the NA of 0.78 is not considered large, we agree with the reviewer that a quantitative description of the relative X and Z contributions is warranted and thus have added this on page 4 and 11. With regards to the variation in incidence angle, we employ a 38cm FL parabolic mirror to focus the beam, starting at a width of ~5mm and ending up with ~100 μ m. Therefore, even with this simplistic approach, the variation of incidence angle would be <1°. As this was raised, we have also added this information to the manuscript on page 11.

In line 133, the statement ‘Nevertheless, the simple presence of these CH2 responses inside the domains shows that the signals from the individual lipids are not cancelling within each pixel, indicating substantial co-directional molecular in-plane ordering at the sub-micron scale.’ requires additional explanation. The presence of CH2 response could also be attributed to the presence of gauche defect (Phys. Rev. Lett., 1987, 59, 1597–1600)

We agree with the reviewer that this statement needs further explanation and thus have modified this section on page 5 to better describe why the observed variations in Fig. 2 suggest a clear in-plane directionality. Furthermore, as these are only suggestive at this stage, we have moved the conclusive statements about the co-directional in-plane ordering to after the rotational analysis on page 7.

In the azimuth angle analysis, what is the physical interpretation of in-plane/out-of-plane component? It should be related to the tensor element of second-order nonlinear susceptibility, as well as the Euler angles (θ, ϕ, φ). The more detailed discussion should be included. For example, for a C_1 symmetry component, all 2nd order susceptibility (and hyperpolarizability)

are allowed, the current argument seems to be too hand-waving, and lack of a solid theoretical foundation.

As we noted above, the analysis is supported by a solid theoretical foundation, and have made appropriate modifications to the manuscript to better highlight the second-order nature of the signals and how they relate to the in-plane and out-of-plane components. Specifically, as the reviewer suggested, these are directly related to the molecular Euler angles defining their directionality.

In line 265, the authors calculated the lipid order and density based on several assumptions. However, the SFG response is the product of the molecular surface density (N), an order parameter (O), and the hyperpolarizability (β). The authors did not consider the hyperpolarizability in their analysis. The order parameter is typically determined by performing different polarization combination experiments in the SFG community. Therefore, it would be beneficial to conduct such experiments and discuss whether the results align with their current analysis.

The reviewer is correct that the SFG response includes all three contributions and, although it is not directly discussed, we can confirm that we did indeed consider the hyperpolarisability in our analysis. As the orientation-independent SFG magnitudes are isolated, the resulting SFG signal is only determined by the number density of the lipid, and the relative amount of cancellation of different hyperpolarisability contributions within each pixel. As the hyperpolarisability is a molecular property, it is thus a constant in these terms and thus can be neglected. We do acknowledge, however, that this differs from the conventional definition of an ‘order parameter’ in SFG and thus have altered this section in the Supplementary Information to emphasise this point and instead define it as an orientational parameter to describe the amount of cancellation from the spatial averaging of the hyperpolarisability within each pixel.

Regarding the determination of the order parameters using different polarisation combinations, we completely agree that this is desirable information. This is generally achieved in spectroscopy measurements under the assumption of isotropic azimuthal symmetry and a specific orientational distribution functional form, often taken to be Gaussian. With our microscope the different polarisation combinations are indeed all accessible. In the study presented here, however, the extraction of the order parameter is highly infeasible. Firstly, there is no symmetry in the system and thus each polarisation combination is contributed to by many more tensor components that can all interfere and, in principle, need to be separated. Secondly, due to the lack of symmetry, it is far from clear what functional form the orientational distribution takes. On inspection of Eq. 4 in the referenced JCP paper, it is clear that the orientational distribution function contains functions for all three Euler angles. In spectroscopic measurements, along with c_{∞} symmetry at the interface, it is often also assumed that the rotation and twist angles are isotropically distributed and thus the only parameters come from the functional form of the tilt angle distribution. Clearly in the systems investigated in this work this is not case and thus the orientational distribution function contains potentially three different functional forms and thus many unknown parameters. Even assuming Gaussian distributions of each Euler angle, six parameters are required and thus the measurement of a minimum of 6 polarisation combinations would be necessary. Furthermore, such analysis would need to extend down to the single pixel level, necessitating exceptionally high data quality. Indeed, we are working towards the extraction of this interesting information, but such experimental analysis requires another significant technical advance to become feasible.

It is also worth noting that the presented analysis in the Supplementary Information already contains several assumptions which render a precise quantitative analysis superfluous, but instead can be used to give a clear indication on the comparison of the extents of in-pixel cancellation between the two lipid mixtures investigated. For this reason, the results from this are given solely as indications (as stated in the manuscript) of differences between the two lipid mixtures, in contrast to the central findings of the work i.e., the in-plane packing structure of the biomembranes and observed enantioselective interactions, which are based on quantitative grounds. Nevertheless, we are convinced that the presented insights in combination with the technical advancement aspect of the paper makes it sufficiently impactful and of interest to a wider community even without the absolute determination of the order parameters.

In line 625, the statement ‘an order parameter, O , that describes the width of the orientational distribution of the molecular transition dipoles’ is incorrect. People usually assume different functions ($f(\theta)$) like exponential decay, Gaussian, or stepwise to describe the orientational distribution. The order parameter is the average of ensemble i.e. $\langle A \rangle = \int_0^\pi A f(\theta) \sin\theta d\theta$, $A = \cos\theta$ or $[\cos]^3 \theta$. See the references (J. Chem. Phys. 21 March 2006; 124 (11): 114705 and Phys. Rev. Lett. 121, 246101). As such, the order parameter cannot provide information about the width of the orientational distribution.

We agree with the reviewer that the wording here is somewhat misleading, but would like to note that the order parameter is clearly highly linked to the orientational distribution and thus also its width. Nevertheless, we have redefined this parameter as noted above and more accurately described its relation to the amount of orientational cancellation.

In the chirality study, people usually use chiral SFG spectroscopy for characterizations of molecular structures at interfaces (Chem. Rev. 2014, 114, 17, 8471–8498). However, there is no any discussion of chiral SFG in the manuscript. If the author performed the chiral SFG experiment, could they get the consistent result as their current analysis? The discussion and references about chiral SFG should be included.

Lastly, there could be two types of chiral signals, one coming from the chirality of the molecules themselves, another coming from a chiral structure arranged by molecules (either chiral or achiral) in a specific way. It seems that the authors could see both of them. It would be helpful if the authors can discuss more about the origin of the signals.

The reviewer raises an interesting point here regarding chiral SFG signals. As they note, there can be two types of chiral signals, namely chiral molecular signals and chiral signals from the mesoscopic structure. In the former case, as these are purely associated with the molecules themselves, they are reflected by chiral hyperpolarisability components and thus are related to the local vibrational modes of chiral centres within the structure (i.e., vibrational modes with local chiral symmetry). In the latter case, the chiral signals arise from the overall lack of symmetry in the molecular packing structure (i.e., C1) and thus can source from non-chiral molecular hyperpolarisability components, however, such signals are typically small. In this work, we are predominantly focussed on determining the lateral lipid packing structure (structural chirality), and thus on the tail groups which do not possess chiral hyperpolarisability contributions. Therefore, for the microscopy studies it makes most sense to probe non-chiral polarisation combinations such as PPP since the desired chiral information can directly be obtained from the spatial mapping of molecular orientations. As the reviewer suggested, our added theoretical description in the Supplementary Information now fully describes the origins of the signals. However, whilst we completely agree with the reviewer that a discussion on

chiral SFG spectroscopic studies of these systems is very interesting, we are convinced that this does not fit within the scope of this work. Furthermore, we believe that, although chiral SFG spectroscopy is clearly a powerful technique, for the samples investigated here, it would only yield very limited additional insight.

Chiral SFG spectroscopy studies, such as those within the referenced review paper above, typically assume c_{∞} symmetry such that polarisation combinations like PSP only contain chiral susceptibility contributions. From the microscopy studies presented here, however, we show that the local symmetry of these lipid membranes is C_1 and, even when the signals are spatially averaged, will deviate from this assumption of c_{∞} due to the size and curvature of the lipid domains. In fact, in our measurements we observed that the spatially averaged spectra clearly change upon sample rotation, showing that this assumption is not entirely valid. Therefore, it is likely that spectroscopic chiral SFG studies include other tensor components that are thought to be zero, thus somewhat mudding the interpretation of their spectra. This emphasises that the investigation of such samples really benefits from SFG microscopy studies where these symmetry assumptions are not necessary.

On the other hand, chiral SFG studies can be powerful for studying molecular chirality. These studies, however, can also be challenging owing to the potential contribution of both chiral and achiral hyperpolarisability components to the chiral susceptibilities. This makes it difficult to separate local and mesoscopic chirality in the sample without any prior knowledge.

Finally, even discounting the achiral susceptibility contributions, chiral SFG study of such lipid systems tend to yield relatively weak signals. This is demonstrated in a recent paper studying pure DPPC monolayers with chiral SHG and SFG where small contributions from the CH₃ SS were observed and assigned to chiral susceptibility components. [J. Chem. Phys. 156, 094704 (2022)] The most significant contribution, however, was observed due to the head-group, arising from a CH₂ adjacent to the chiral centre. It is worth noting, however, that this study also assumed an isotropic interface and thus may have additional contributions that are not considered. Furthermore, whilst the observation of apparent chiral contributions in the CH₃ SS does indicate macroscopic chirality (since the CH₃ group is intrinsically achiral), the interpretation of these signals is far from trivial, with even the source of the chiral contributions from this band being complicated and potentially involving intermolecular couplings. [J. Phys. Chem. B 2021, 125, 43, 12072–12081] For this reason, the authors use prior knowledge of the packing curvature to justify their observations rather than the other way around. This demonstrates that only limited insight into the packing structure can be gained from such spectroscopic measurements and, again, reiterates the need for microscopic studies.

As seen above, including such a discussion in the paper would be quite involved and thus substantially lengthen the manuscript and deviate from the central points being made. Specifically, we are predominantly focussed on spatial imaging of these heterogeneous systems. Nevertheless, we plan to include such a detailed discussion as part of our future in-depth description of the methodology in a more specialised journal.

In summary, the paper presents the new methodology and data for characterizing the lipid domain with mixed chirality. The conclusions reached are broadly in line with earlier work. The discussion in some places is lack of necessary details to establish a solid foundation to appreciate the data obtained from this advanced technique (see above). A more specialized journal might allow for a more rigorous discussion.

Once again, we thank the reviewer for providing us with comments and suggestions based on their expertise. As detailed above, we have made several modifications to the manuscript and demonstrated in our response that the presented results are indeed based on a solid analysis of the second-order signals. We therefore hope that the reviewer's concerns are now sufficiently satisfied such that they can appreciate the presented data. Finally, as mentioned earlier and as is suggested by the reviewer here, a rigorous discussion of the underlying theoretical considerations, whilst not being appropriate for the readability of this paper, is planned for a more specialised journal soon such that anyone within the SFG field can access the advanced technical details behind the method.

Reviewer #3

In this manuscript, Fellows et al. describe a new microscopy technique that allow the visualization and study of lipid rafts in lipid monolayers. The authors show that by probing the C-H stretching vibrations of DPPC domains it is possible to resolve the microscopic structure of lipid monolayers. The paper is well-written and the results are well-supported by the experiments. I recommend its publication in Nat. Commun. after addressing the following comments:

We would like to sincerely thank the reviewer for taking the time to review our manuscript and provide us with detailed feedback. For each of the point raised, we have made alterations to the paper as detailed in our responses below.

- My main concern is related to the clarity of the in-plane molecular directionality. While the out-of-plane directionality is clearly explained and represented, the in-plane directionality of individual molecules, which is central to this work, is not clearly explained and it is hard to visualize despite the expansion present in Figure 1a. It is not clear what the black and blue arrow indicates. Since the in-plane directionality derives from the chirality, a 3D representation of the phospholipid molecule would help clarify this concept and make the manuscript more accessible to a broader readership.

We agree with the reviewer that the visualisation of the molecular directionality is central to our work and that the way it was represented in Figure 1 was not necessarily clear. As such, we have modified the representation given in Figure 1 as per the reviewer's suggestion to include a more 3D representation of the molecule. Additionally, we have sought to clarify the description of the molecular direction through added labelling and better clarity of wording in the figure caption.

- The developed microcopy technique seems very powerful to characterize lipid monolayers but lipid bilayers are never mentioned throughout the manuscript. This is a limitation that should be mentioned and discussed. If the technique is applicable only to monolayers, this should be reflected in the title of the paper.

This is a very important point that the reviewer has raised about the wider applicability of the technique. Although we initially aimed to solely focus on the ability of the technique for probing the monolayer samples being investigated, hoping that the results and analysis would be sufficient to infer its wider applicability, we agree that an explicit mentioning of the capabilities of the technique is beneficial. Therefore, as the reviewer suggested, we have added to the discussion of the technique and its outlook on page 10 to specifically address bilayer samples due to their relevance to this work. In short, while there is no fundamental reason bilayers could not also be studied, this represents an additional experimental challenge that we are currently working towards.

- The DPPC/DOPC ratio used to obtain lipid rafts should be mentioned at least once in the "Results" section.

We thank the reviewer for drawing our attention to this point. Whilst a full description of the sample systems being investigated, including the lipid ratio, is given in the methods section, we agree that it is also useful to present this information when first discussing the system being probed. Therefore, whilst it is not specifically in the "Results" section, we have added to this information on page 4 in the paragraph immediately preceding the results.

- The observation that DPPC is present both in liquid-condensed (LC) and liquid expanded (LE) phases is very interesting. Would it be possible to estimate their proportion? If yes, given a specific DPPC/POPC ratio, this would allow to study and what factors influence the formation of lipid rafts.

While we completely agree that the derivation of this information would be highly desirable, leading to our analysis of the SFG images given in the “Domain Density and Orientational Order Calculations” section of the Supplementary Information, unfortunately we don’t believe such parameters can be extracted from the presented data. This is predominantly due to the convoluted nature of SFG signals which probe both molecular density, order, and orientation. In our measurements, by rotationally characterising the signals, we can generate an effective SFG amplitude for each pixel, independent on the specific orientation of the molecule, but cannot distinguish between alterations to the molecular density or the relative order as both will change the SFG amplitude. We are currently working on improvements to this analysis of lipid domains by measuring similar samples with both lipids being protonated, thus both giving SFG spectra that don’t entirely overlap. Based on this, they are potentially separable. With this data, along with accurate surface pressure isotherm measurements and appropriate assumptions, it could become possible to access such quantities. However, due to complexity of such studies we are currently incapable of providing the desired results. Nevertheless, we hope to be able to determine these in the future.

- The findings summarized with the sentence “We find that both mixtures form domains based on mirror-symmetric spiral structures, but represent different regions within such motifs. This breaks their mirror symmetry” should be better described and discussed. The authors should explain better the meaning of breaking the symmetry and from what evidence these findings are derived.

We thank the reviewer for pointing out the lack of clarity in this symmetry argument and agree that a more detailed explanation of this point is warranted. As such, we have extended the discussion of the molecular packing and breaking in mirror symmetry on page 9 to more clearly describe the differences between the two types of domains and how this relates to enantioselectivity.

REVIEWER COMMENTS

Reviewer #1 (Remarks to the Author):

The authors have addressed our concern regarding distinguishing between LC/LE and LO/LD systems. The author has argued the benefits of using v-SFG compared to other microscopy techniques, as well as the unique capability of distinguishing directionality. However, the argument regarding presence of fluorescence tag interfering with true measurements is controversial as much of our "common" knowledge of how lipid molecules self-assemble has been deduced from experiments utilizing fluorescent probes. This also does not deal with Brewster angle microscopy, which is also a non-probe technique. The tilt directionality in these optical probes is determined $\pm \pi$ so the SFG provides only a marginal improvement. The speculation origin behind enantiomeric selectivity has been laid out with a thermodynamic argument, however a more quantitative description would be preferred. As such, the claims in the abstract should be softened.

Reviewer #2 (Remarks to the Author):

The authors have addressed quite a few of my questions well. In particular, I applaud the authors efforts in addressing my main concern of the lack of solid theoretical foundation. The text they included in SI is just sufficient enough to put the correct amount of theoretical context for their experimental work. There are still a few parts of my questions need more clarification. Also, I appreciate if the authors could copy/paste their changes to the text or point out the location of changes in the manuscript in their next round of revision.

In their reply point 1, The new sentences are "In any case, we can further confirm that the phase of our presented spectrum is correct by considering the PPP spectra of SAMs on gold (which is effectively only the ZZZ contribution) and the SSP spectrum of a lipid monolayer (which probes YYZ=XXZ in azimuthally symmetric systems). The ZZZ contribution appears solely as dips for the three methyl resonances (given that gold is well-known to have a positive phase for its NR contribution) whereas the YYZ (XXZ) appears as dips for the SS and FR, but a peak for the AS. Therefore, as the spatially averaged PPP spectrum is effectively given by ZZZ - XXZ, this leads to cancellation of intensity for the SS and FR and a strong negative band for the AS ". This analysis is not very clear to me. It would be better if the authors could provide the data (ssp spectrum) in SI and/or mention the literature in their manuscript."

In their reply point 2, this reply is oversimplifying. The new sentence is "As depicted in Fig. 1a, these tails are expected to be in a largely upright configuration with their terminal CH3 groups all pointing upwards. For this structure, the vSFG responses should be dominated by the CH3 groups and yield a positive contribution from the symmetric stretch (SS) and negative band from both antisymmetric (AS) modes". Like what the authors mentioned in the reply, the positive/negative sign in the ppp spectrum also depends on the SFG geometry, Fresnel factor, and the cancelation of different tensor elements. Only the molecular structure in Fig. 1a cannot determine the sign of the peaks of symmetric stretch and antisymmetric stretch. It should not be difficult for the authors to implement an Euler rotation analysis and put their argument into a solid footing, instead of claiming a "well-defined link" here.

In their reply point 3, I appreciate the authors included the additional information to be quantitative. Very helpful and transparent. I am just confused about the sentence on page 10 of the main text that says, "the electric fields are directed with ~58% in-plane and ~42% out-of-plane contributions." The in-plane and out-of-plane contributions of electric fields should be $E_p \cos(36^\circ)$ and $E_p \sin(36^\circ)$, where E_p is the electric field in the p polarization and 36° is the incident angle relative to the surface normal.

In their reply point 4, the updated sentence is "Interestingly, at the CH₂ frequencies, many of the domains also show a split positive/negative character. While the presence of CH₂ SFG signals in such monolayers is traditionally interpreted as arising from conformational (Gauche) defects in the alkyl chains, these amplitude variations across the domains are clearly incompatible with the random nature of defects. Instead, having both positive and negative contributions that are locally correlated suggests slowly varying in-plane directionality." But I am still confused the interpretation. If all the lipids are trans conformation without gauche effect, the transition dipole moment of CH₂ should be cancelled out even within a single molecule, resulting in the negligibly in-plane/out-plane contribution of CH₂ response. The argument of amplitude variation could suggest a gauche defect depending on the mesoscopic packing of the systems too. The second point is that the sample is DPPC:POPC = 4:1. It has already shown that POPC lipid has a strong CH₂ SFG response (J. Phys. Chem. B 2020, 124, 25, 5246–5250 and J. Chem. Phys. 156, 234706 (2022)). How do they exclude the possibility that the CH₂ response is from the POPC? As the authors indicated that these statements are only suggestive instead of conclusive at this stage, a simple improvement is be open to these other possibilities in their discussion.

In their last reply point about chiral SFG signal contribution, I agree with the authors analysis. My original point was that there could be a microscopic chiral SFG signal encoded in the systems which is beyond the spatial resolution of the microscope. Thus, chiral SFG could provide complementary chirality information at a different length scale from the mesoscale obtained from their current imaging method. I believe this discussion will strengthen the work, but it seems the authors were defensive about this suggestion.

Reviewer #3 (Remarks to the Author):

The authors have responded effectively to my comments/concerns

Reviewer #4 (Remarks to the Author):

Reviewer #5 (Remarks to the Author):

Reviewer #6 (Remarks to the Author):

Response to Reviewers' Comments

Reviewer #1

The authors have addressed our concern regarding distinguishing between LC/LE and LO/LD systems. The author has argued the benefits of using v-SFG compared to other microscopy techniques, as well as the unique capability of distinguishing directionality. However, the argument regarding presence of fluorescence tag interfering with true measurements is controversial as much of our “common” knowledge of how lipid molecules self-assemble has been deduced from experiments utilizing fluorescent probes. This also does not deal with Brewster angle microscopy, which is also a non-probe technique. The tilt directionality in these optical probes is determined $\pm \pi$ so the SFG provides only a marginal improvement. The speculation origin behind enantiomeric selectivity has been laid out with a thermodynamic argument, however a more quantitative description would be preferred. As such, the claims in the abstract should be softened.

Once again, we would like to thank the reviewer for taking the time to review our manuscript and the revisions made from their previous comments. We are pleased that our changes based on the justification of studying the interactions of lipids within LC/LE systems has satisfied their concerns. With regards to their remaining two comments, as we argue below, we do not believe any further changes to the manuscript are warranted.

As per their request in the previous round of review, we made changes to the manuscript to emphasise the importance of previous PFM studies and contextualise our investigation using SFG microscopy. For this, we highlighted the specific importance of obtaining absolute directionality which, as the reviewer notes, is not available through other microscopy techniques. Furthermore, we noted the complications of adding fluorescent probes to the system, both from the perspective of potential structural perturbations, and due to the unclear relation between observed directionality with that of the lipids. In their latest comments, the reviewer suggests that the information accessible through SFG microscopy only represents a “marginal improvement” to that available from polarised fluorescence microscopy (PFM). We firmly disagree with this statement. Even disregarding the fact that PFM probes the tag orientation $\pm\pi$, not that of the lipid, not accessing absolute directionality leaves the packing structure somewhat ill-defined. Therefore, combining this with the fact that the lipid directionality is not directly probed means the obtained information on the specific molecular packing structure clearly remains ambiguous. Furthermore, while perturbations induced by the tag may be controversial, clearly not having a tag, as in SFG microscopy, removes this controversy entirely. As a final note, SFG microscopy is not just a structural imaging technique, but also gains access to vibrational spectra, thus, in principle, allowing for molecular recognition and the analysis of molecular compositions and local environments. While we agree that a substantial portion of our current view on lipid membrane structure comes from utilising fluorescent probes, thus highlighting their importance, we do not see how using a new technique which can probe entirely unperturbed systems and gain previously unattainable, but crucial insight into such systems would not be of significant value to the field. On the contrary, there is clearly potential in SFG microscopy to specifically address the above controversies in PFM by comparing the molecular directions of the molecules with and without a tag, as well as between the molecules and the tag itself. In that sense, we see that the fluorescence microscopy

field could in fact highly benefit from the introduction of SFG microscopy as a complimentary imaging technique.

With regards to the reviewer's last comment about the speculative connection to the origin of enantiomeric selectivity, as they note, our previous changes to the manuscript put greater emphasis on the thermodynamic comparison between the two different chiral mixtures, linking these to the clear differences between the structures of their formed domains. As per the previous request of the reviewer, we also made our associated statements linking these observations to enantiomeric selectivity in biomembranes less strong and more suggestive. While we completely agree that a more quantitative thermodynamic comparison between the two systems would be highly desirable and informative, this is not possible from our measurements, but we nevertheless maintain our conclusions about the undeniable enantioselective interactions existing within the mixed chirality membranes. Furthermore, we would like to highlight that our statements in the abstract and discussion essentially read as: the clear enantioselectivity which ultimately governs the mesoscopic structure of the membrane "may be relevant" / "could be a relevant factor" in the evolution of homochirality. We therefore believe that these statements are already sufficiently softened and correctly represent the wider implications of our findings, and are thus not overselling the conclusions.

Reviewer #2

The authors have addressed quite a few of my questions well. In particular, I applaud the authors efforts in addressing my main concern of the lack of solid theoretical foundation. The text they included in SI is just sufficient enough to put the correct amount of theoretical context for their experimental work. There are still a few parts of my questions need more clarification. Also, I appreciate if the authors could copy/paste their changes to the text or point out the location of changes in the manuscript in their next round of revision.

We are sincerely grateful to the reviewer for continuing to provide us with feedback on our manuscript based on their expertise, and are glad that our theoretical additions have broadly satisfied their concerns. For each of the further comments detailed below, we have responded and made appropriate changes to the manuscript. Regarding our previous changes, we deeply apologise if these were not clear enough in the previous round of review. With our new revisions, we have made additional efforts to specifically describe the changes and highlight where they appear in the manuscript in our responses below.

In their reply point 1, The new sentences are “In any case, we can further confirm that the phase of our presented spectrum is correct by considering the PPP spectra of SAMs on gold (which is effectively only the ZZZ contribution) and the SSP spectrum of a lipid monolayer (which probes $YYZ=XXZ$ in azimuthally symmetric systems). The ZZZ contribution appears solely as dips for the three methyl resonances (given that gold is well-known to have a positive phase for its NR contribution) whereas the YYZ (XXZ) appears as dips for the SS and FR, but a peak for the AS. Therefore, as the spatially averaged PPP spectrum is effectively given by $ZZZ - XXZ$, this leads to cancellation of intensity for the SS and FR and a strong negative band for the AS “. This analysis is not very clear to me. It would be better if the authors could provide the data (ssp spectrum) in SI and/or mention the literature in their manuscript.”

As this point was raised by the reviewer, we have added a new section to pages 6-8 in the SI entitled “Appearance of the Spectra and Out-of-plane Molecular Orientation” to specifically address it. This includes the integrated SSP spectrum and literature references, as requested, and we have further utilised the SSP spectrum to deconvolute the presented PPP spectrum into its two surviving tensor elements to discuss the appearance of the main resonances in the overall PPP spectrum.

In their reply point 2, this reply is oversimplifying. The new sentence is “As depicted in Fig. 1a, these tails are expected to be in a largely upright configuration with their terminal CH₃ groups all pointing upwards. For this structure, the vSFG responses should be dominated by the CH₃ groups and yield a positive contribution from the symmetric stretch (SS) and negative band from both antisymmetric (AS) modes”. Like what the authors mentioned in the reply, the positive/negative sign in the ppp spectrum also depends on the SFG geometry, Fresnel factor, and the cancelation of different tensor elements. Only the molecular structure in Fig. 1a cannot determine the sign of the peaks of symmetric stretch and antisymmetric stretch. It should not be difficult for the authors to implement an Euler rotation analysis and put their argument into a solid footing, instead of claiming a “well-defined link” here.

We agree that perhaps the current statement is too oversimplifying and warrants a more detailed discussion. As this is also linked to the general appearance of the main resonances in the PPP spectrum, we have combined this within the added section on page 6-8 in the SI in response to the comment above which includes the SSP spectrum. Furthermore, we have also added the

specific Euler transformation we employ for our analysis in the theoretical section in the SI on page 5 (Eq. 3), and refer to this when discussing the sign of the signals in terms of the molecular tilt angle.

In this added section, we now specifically address the expected signs the CH₃ resonances in the overall PPP response in relation to the molecular tilt angle, incident angle, and Fresnel factors, and how the multiple tensor components make such an analysis non-trivial. Nevertheless, through their theoretical definitions based on the Euler transformation, we demonstrate how the SS must be positive and the AS negative, as mentioned in the main text. We also then discuss the overall appearance of the observed PPP spectrum and, using the SSP spectrum, demonstrate how the observations fit perfectly well with a ‘pointing up’ orientation. This new SI section is now referred to immediately following the statement on page 5 in the main text: “For this structure, the vSFG responses should be dominated...”, where we have also added “(with a 36° incidence angle)” to emphasise that this expectation is related to the specific incidence angle being employed.

In their reply point 3, I appreciate the authors included the additional information to be quantitative. Very helpful and transparent. I am just confused about the sentence on page 10 of the main text that says, “the electric fields are directed with ~58% in-plane and ~42% out-of-plane contributions.” The in-plane and out-of-plane contributions of electric fields should be $E_p \cos(36^\circ)$ and $E_p \sin(36^\circ)$, where E_p is the electric field in the p polarization and 36° is the incident angle relative to the surface normal.

The reviewer is completely correct that the in-plane and out-of-plane contributions are given by $E_{ip} = E_p \cos(36^\circ)$ and $E_{op} = E_p \sin(36^\circ)$, respectively. It is important to note, however, that the proportion of each component cannot be simply taken as cos/sin of the angle as this is, instead the projection of the field onto the respective axis (i.e., the two do not sum to unity). Instead, the relative proportions of the two components are given by:

$$\frac{E_{ip}}{E_{ip} + E_{op}} = \frac{\cos(36^\circ)}{\cos(36^\circ) + \sin(36^\circ)} = 0.579$$
$$\frac{E_{op}}{E_{ip} + E_{op}} = \frac{\sin(36^\circ)}{\cos(36^\circ) + \sin(36^\circ)} = 0.421$$

As our added information led to this confusion, we have now modified the text on page 11-12 by explicitly giving these equations to show how these figures are derived.

In their reply point 4, the updated sentence is “Interestingly, at the CH₂ frequencies, many of the domains also show a split positive/negative character. While the presence of CH₂ SFG signals in such monolayers is traditionally interpreted as arising from conformational (Gauche) defects in the alkyl chains, these amplitude variations across the domains are clearly incompatible with the random nature of defects. Instead, having both positive and negative contributions that are locally correlated suggests slowly varying in-plane directionality.” But I am still confused the interpretation. If all the lipids are trans conformation without gauche effect, the transition dipole moment of CH₂ should be cancelled out even within a single molecule, resulting in the negligibly in-plane/out-plane contribution of CH₂ response. The argument of amplitude variation could suggest a gauche defect depending on the mesoscopic packing of the systems too. The second point is that the sample is DPPC:POPC = 4:1. It has already shown that POPC lipid has a strong CH₂ SFG response (J. Phys. Chem. B 2020, 124,

25, 5246–5250 and *J. Chem. Phys.* 156, 234706 (2022)). How do they exclude the possibility that the CH₂ response is from the POPC? As the authors indicated that these statements are only suggestive instead of conclusive at this stage, a simple improvement is to be open to these other possibilities in their discussion.

The reviewer is completely correct with their analysis about the CH₂ signals in relation to the molecular conformation. While an all-trans conformation fundamentally cannot give rise to CH₂ signals, a Gauche conformation can. Our view underlying this statement thus rests on the specific terminology used. Describing the source of signals which are so clearly structurally correlated (in-plane directions) as ‘defects’ seems to be a misnomer. We have thus altered our statement on page 5 to better describe the structural implications of the observed signals and how they are not compatible with the classic view of ‘defects’, but rather must be spatially correlated and, importantly, vary in directionality across the domains.

With regards to their second point, while the reviewer is correct that POPC has previously been found to have strong signals at the CH₂ frequencies, in our work presented here the POPC being used is fully deuterated, as we note on page 4 at the end of the introduction and page 11 in the methods section. Therefore, no C-H signals can possibly arise from POPC. As this was raised, we have emphasised that we are only probing DPPC, not POPC on page 5.

In their last reply point about chiral SFG signal contribution, I agree with the authors analysis. My original point was that there could be a microscopic chiral SFG signal encoded in the systems which is beyond the spatial resolution of the microscope. Thus, chiral SFG could provide complementary chirality information at a different length scale from the mesoscale obtained from their current imaging method. I believe this discussion will strengthen the work, but it seems the authors were defensive about this suggestion.

As the reviewer has reiterated their suggestion for a discussion on chiral SFG studies to complement the mesoscopic observations from SFG microscopy, we have added a brief comment on this in the outlook on page 10, highlighting that it could indeed be interesting to probe such tensor elements. As we pointed out in our previous response, however, such measurements would not directly address the main questions about the packing structure investigated in this work, but would rather represent complementary information at a different length-scale (as the reviewer noted). We therefore believe that adding a full, detailed discussion would not be appropriate for this particular paper and could distract the reader from the main conclusions.

Reviewer #3

The authors have responded effectively to my comments/concerns.

We thank the reviewer for giving up more of their time to review our revisions and are delighted that all of their comments and concerns have been satisfied.

REVIEWERS' COMMENTS

Reviewer #1 (Remarks to the Author):

I would still claim that the SFG method reinforces existing work that shows the spiral orientation of the tilt director in DPPC semi-crystalline domains, and this has been well established by polarized fluorescence (which requires a probe) as well as Brewster angle microscopy, which does not require a probe. The evolutionary impact of the nature of the tilt orientation defect for left and right handed molecules is still speculative. However, the paper should be published to inspire more discussion on this topic.

Reviewer #2 (Remarks to the Author):

it is good for being accepted

Reviewer #4 (Remarks to the Author):

The authors have addressed my concerns.

Reviewer #6 (Remarks to the Author):
